# Generalization Bounds for Discrete Diffusion: Statistical Advantage of Masking

Zixuan Zhang[1]   Hengyu Fu[2]   Zhuoran Yang[3]   Mengdi Wang[4]   Tuo Zhao[1]   Minshuo Chen[5]

## Abstract

Discrete diffusion models have recently emerged as a compelling alternative for language generation, enabling efficient non-autoregressive sampling while achieving strong empirical performance. A key design choice in discrete diffusion—absent in most continuous diffusion formulations—is the forward corruption kernel, with masked/absorbing corruption now dominating practice. Despite this empirical preference, there is limited statistical theory explaining when and why masking should outperform alternative kernels such as uniform replacement. In this paper, we take a step toward closing this gap from a statistical learning perspective. Our analysis establishes generalization bounds and, through an explicit comparison across different forward corruption kernels, reveals a central advantage of masking: it scales with the effective data support rather than the full ambient state space, thereby mitigating the curse of state space cardinality. We further derive structure-aware refinements that capture how concentration and sparsity in real sequential data sharpen the sample complexities. Together, these results offer a principled explanation for the empirical strength of masked diffusion and provide guidance for forward-kernel design in discrete generative modeling.

[1]School of Industrial and Systems Engineering, Georgia Institute of Technology, Atlanta, GA, USA [2]Department of Computer Sciences, University of California, Berkeley, USA [3]Department of Statistics and Data Science & Department of Computer Science, Yale University, New Haven, USA [4]Department of Electrical and Computer Engineering, Princeton University, Princeton, USA [5]Department of Industrial Engineering and Management Sciences, Northwestern University, Evanston, USA. Correspondence to: Zixuan Zhang <zzhang3105@gatech.edu>.

*Proceedings of the 43rd International Conference on Machine Learning*, Seoul, South Korea. PMLR 306, 2026. Copyright 2026 by the author(s).

## 1. Introduction

Diffusion models have established themselves as one of the most powerful paradigms in generative modeling (Ho et al., 2020; Song et al., 2020b; Yang et al., 2023; Nichol & Dhariwal, 2021). Through iterative denoising processes, these models have achieved state-of-the-art results across diverse domains, such as high-resolution image synthesis (Dhariwal & Nichol, 2021; Rombach et al., 2022; Karras et al., 2022), sequential data modeling (Kong et al., 2020; Tashiro et al., 2021), language modeling (Li et al., 2022; Lou et al., 2023; Nie et al., 2025), computational biology (Watson et al., 2023; Corso et al., 2022), and robotics (Chi et al., 2025). More recently, discrete diffusion models have emerged as a compelling alternative for language generation. These models leverage discrete token-level forward–backward dynamics and have demonstrated not only strong empirical performance but also the ability to generate language sequences more efficiently than traditional autoregressive approaches (Austin et al., 2021; He et al., 2023; Lou et al., 2023; Sahoo et al., 2024; Nie et al., 2025; Ye et al., 2025).

Continuous diffusion models, which first brought diffusion-based generative modeling into prominence, operate within a constrained design space. In continuous domains, the forward process is almost always instantiated as Gaussian noise injection (Ho et al., 2020; Song et al., 2020b). While this approach offers mathematical convenience and natural interpretability, it significantly constrains the ability to tailor the corruption process to specific data structures. Consequently, research in continuous diffusion has predominantly focused on enhancing samplers, network architectures, and guidance mechanisms rather than innovating on the forward-process design itself (Song et al., 2020a; Nichol & Dhariwal, 2021; Karras et al., 2022).

In contrast, discrete diffusion models introduce a substantially richer design space through their forward corruption processes (Hoogeboom et al., 2021; Austin et al., 2021; Campbell et al., 2022). Rather than employing Gaussian noise injection, these models simulate Markov jumps to corrupt clean data according to pre-determined transition kernels (Campbell et al., 2022; Chen & Ying, 2024). Several such kernels have been extensively explored in the litera-

ture. A canonical example is *uniform replacement*, where each token is randomly substituted with a uniformly drawn vocabulary element at a time-dependent rate (Austin et al., 2021; Hoogeboom et al., 2021). Another widely adopted approach is the *masked (absorbing) process*, wherein tokens are progressively replaced by a special mask symbol, causing the chain to monotonically converge toward a fully masked sequence (He et al., 2023; Sahoo et al., 2024; Ou et al., 2024). Empirically, this masked/absorbing corruption has established itself as the default mechanism in state-of-the-art discrete diffusion models for language generation, consistently delivering superior generation quality and more efficient sampling compared to uniform replacement approaches (He et al., 2023; Sahoo et al., 2024; Nie et al., 2025; Ye et al., 2025).

Despite the practical dominance of masked diffusion, the statistical foundations underlying its superiority remain poorly understood. Existing work on discrete diffusion has primarily focused on *modeling and algorithms*—including novel parameterizations of trainable parameters, advanced training objectives, and efficient decoding schemes—while largely treating the forward transition kernel as a design choice justified merely by empirical performance (Austin et al., 2021; He et al., 2023; Lou et al., 2023; Sahoo et al., 2024; Campbell et al., 2022; Ou et al., 2024). Concurrently, the continuous diffusion literature has developed substantial theoretical frameworks addressing training and sampling processes, with particular emphasis on generalization bounds and sampling efficiency analyses (Chen et al., 2022; Benton et al., 2023; Li et al., 2024; Stéphanovitch et al., 2025; Yakovlev & Puchkin, 2025; Zhu et al., 2023; Chen et al., 2023; Oko et al., 2023; Zhang et al., 2024a; Dou et al., 2024).

However, these theoretical advances in continuous diffusion cannot be directly transferred to discrete settings: the data domain is fundamentally discrete, the corruption mechanism departs from Gaussian noise, and most significantly, the forward transition kernels enable qualitatively different approaches to information destruction (e.g., masking versus uniform replacement). Consequently, current theory provides limited guidance on the following fundamental question:

*Why do masked diffusion models yield stronger performance, and more broadly, what principles govern the generalization ability of discrete diffusion models?*

**Our contributions** In this paper, we address this fundamental question by establishing a comprehensive sample-complexity theory for discrete diffusion models that explicitly characterizes the role of the forward corruption process. Our main contributions are threefold:

● **General statistical framework**. We develop a unified theoretical framework for analyzing distribution estimation in discrete diffusion models with general forward processes (Section 3.1). Our analysis reveals that the range of probability ratios induced by the forward process constitutes the critical factor governing sample complexity and generalization performance.

● **Sample complexity comparison**. We prove that absorbing processes achieve sample complexity scaling with the support size of the underlying data distribution, while uniform processes necessarily scale with the dimensionality of the entire state space (Section 3.2). To our knowledge, this represents the first result theoretically distinguishing the statistical efficiency of different forward processes in discrete diffusion.

● **Adaptivity analysis**. We further develop refined bounds demonstrating how both processes can adapt to distributional structure through distinct mechanisms: uniform processes leverage uniformity of probability mass, while absorbing processes exploit sparsity in valid token completions (Section 4).

Together, these results establish a rigorous theoretical foundation for understanding why masked diffusion has emerged as the dominant approach in practice, and offer principled guidelines for designing effective discrete diffusion models across structured discrete domains. In addition, we highlight the technical challenge underlying our analysis: establishing these results requires controlling the statistical estimation error along the entire continuous-time corruption trajectory, where the support and probability range of the corrupted distribution evolve with time. Tracking how each forward kernel reshapes the support along its corruption path precisely reveals the separation between absorbing and uniform processes.

**Related work** Several empirical studies have demonstrated the superior performance of masked diffusion language models, establishing this approach as the dominant design choice. Notable examples include DiffusionBERT (He et al., 2023), MDLM (Sahoo et al., 2024), and SEDD (Lou et al., 2023). While alternative formulations such as uniform replacement are frequently used as baseline corruption mechanisms in discrete diffusion frameworks (Austin et al., 2021; Campbell et al., 2022), masked diffusion models consistently outperform these alternatives (He et al., 2023; Sahoo et al., 2024; Nie et al., 2025; Ye et al., 2025). On the uniform-state side, recent work leverages a duality between uniform-state and Gaussian diffusion to improve training and sampling (Sahoo et al., 2025; Deschenaux et al., 2026); these advances operate at the algorithmic level and

are complementary to the kernel-level statistical comparison we pursue. Recently, diffusion-based language generation has been successfully scaled to large models, demonstrating competitive performance with favorable sampling efficiency (Nie et al., 2025; Ye et al., 2025).

The theoretical understanding of discrete diffusion remains relatively nascent, focusing primarily on two aspects: (i) mechanistic characterizations of discrete diffusion models and (ii) sampling guarantees for well-trained diffusion networks. Campbell et al. (2022); Benton et al. (2024) analyze discrete diffusion dynamics within a continuous-time (Markov process) framework, while Ou et al. (2024) characterize the forward transition kernel in masked/absorbing diffusion by linking it to conditional distributions of clean data. Sampling complexity analyses have been established by Chen & Ying (2024); Zhang et al. (2024b). Chen et al. (2025) investigate optimal sampling schedules for masked diffusion models. Moreover, Liang et al. (2026) establish convergence guarantees for absorbing samplers at rates linear in the dimension, improving over the uniform kernel, and Huang et al. (2025) show the number of inference steps for masked diffusion can be logarithmic in target accuracy. These analyses concern sampling under a given score, which drives the reverse process, rather than how it is estimated from data.

From a statistical learning perspective, Wakasugi & Suzuki (2026); Srikanth et al. (2025) develop statistical complexity bounds. Notably, Wakasugi & Suzuki (2026) prove state size independent bounds by assuming the existence of a continuous embedding of the state space and scores bounded away from zero. These results provide important foundations, but they offer limited insight into how the *choice of forward transition kernel* governs generalization and sample complexity. Moreover, their assumptions are incompatible with the absorbing kernel, which is non-reversible and naturally admits zero scores for invalid completions. Our work fills this gap by comparing masking and uniform forward processes in the original discrete state space under minimal assumptions on the data distribution.

**Notation**: Given an integer $d \in \mathbb{N}^+$, we denote $[d] = \{1, \ldots, d\}$. Given a vector $v \in \mathbb{R}^d$, we denote $v^i$ as the $i$-th entry of $v$, and denote $v^{-i} = \{v_j\}_{j \in [d] \setminus \{i\}}$ as a vector in $\mathbb{R}^{d-1}$ that only excludes $v^i$. Given a matrix $Q$, we denote $Q(i, j)$ as the $(i, j)$-th entry of $Q$. For an index set $\mathcal{A} \subseteq [d]$, denote $v^{\mathcal{A}} = \{v^i\}_{i \in \mathcal{A}}$.

## 2. Discrete Diffusion Model

We consider discrete sequential data $X \in \mathcal{X}$ drawn from an underlying distribution $P_{\text{data}}$. The state space $\mathcal{X} = [K]^d =$ $\{1, 2, \ldots, K\}^d$ consists of sequences of length $d$, where each position takes one of $K$ possible values. Note that the cardinality of this space, $|\mathcal{X}| = K^d$, grows exponentially with the sequence length. We represent $P_{\text{data}}$ as a probability vector $p_{\text{data}}$, where each entry $p_{\text{data}}(x) = \mathbb{P}_{\text{data}}(X = x)$ denotes the probability of observing sequence $x \in \mathcal{X}$.

### 2.1. Discrete Diffusion Models

Given a dataset consisting of $n$ i.i.d. samples $\{x^{(1)}, \ldots, x^{(n)}\}$ from $P_{\text{data}}$, our goal is to estimate this unknown distribution and develop efficient sampling procedures. Discrete diffusion models achieve this by employing coupled forward and backward Markov processes, analogous to their continuous counterparts that use Gaussian noise.

**Forward Process on Discrete Support**    To corrupt the discrete data distribution $P_{\text{data}}$, we construct a continuous-time Markov process governed by a time-dependent transition matrix $Q_t$ for $t \in [0, T]$, where $T$ denotes a sufficiently large terminal time. At any time $t$, the corrupted distribution is represented by $P_t$ with corresponding probability vector $p_t$. The forward evolution follows:

$$\frac{\mathrm{d}p_t}{\mathrm{d}t} = Q_t p_t \quad \text{with} \quad p_0 = p_{\text{data}} \quad \text{and} \quad t \in [0, T]. \quad (1)$$

The resulting terminal distribution $P_T$ depends on the specific choice of $Q_t$. Unlike continuous diffusion models, the discrete setting permits greater flexibility in designing the transition matrix $Q_t$, provided that a well-defined stationary distribution $P_\infty$ exists as $T \to \infty$.

**Backward Process on Discrete Support**    The forward process in (1) induces a corresponding backward process that reverses the corruption:

$$\frac{\mathrm{d}\widetilde{p}_t}{\mathrm{d}t} = \widetilde{Q}_t \widetilde{p}_t \quad \text{with} \quad \widetilde{P}_0 = P_T \quad \text{and}$$

$$\widetilde{Q}_t(x, y) = \begin{cases} \frac{p_{T-t}(x)}{p_{T-t}(y)} Q_{T-t}(y, x) & \text{if } x \neq y, \\ -\sum_{l \neq x} \frac{p_{T-t}(l)}{p_{T-t}(x)} Q_{T-t}(x, l) & \text{if } x = y. \end{cases} \quad (2)$$

Here $\widetilde{p}_t$ is identical to $p_{T-t}$, representing a time reversal of the forward process. Notably, the backward transition matrix $\widetilde{Q}_t$ depends not only on $Q_t$ but also on the ratio $p_{T-t}(x)/p_{T-t}(y)$, which we term the *discrete score function*—a quantity of central importance in our analysis.

Since this probability ratio is unknown in practice, we must approximate it with an estimator $s_t(x, y)$. Additionally, we replace the unknown initial distribution $P_T$ with the stationary distribution $P_\infty$. This yields a practically implementable

backward process:

$$\frac{\mathrm{d}\widehat{p}_t}{\mathrm{d}t} = \widehat{Q}_t\widehat{p}_t \quad \text{with} \quad \widehat{P}_0 = P_\infty \quad \text{and} \tag{3}$$

$$\widehat{Q}_t(x,y) = \begin{cases} s_{T-t}(x,y)Q_{T-t}(y,x) & \text{if } x \neq y, \\ -\sum_{l \neq x} s_{T-t}(l,x)Q_{T-t}(x,l) & \text{if } x = y. \end{cases}$$

**Discrete Score Matching** Estimating the discrete score function parallels score estimation in continuous diffusion models, though with key differences in methodology. Rather than employing a quadratic loss function, Lou et al. (2023) introduced a score entropy approach that preserves the positivity constraint inherent to probability ratios. This loss function takes the form:

$$\underset{s_t}{\operatorname{argmin}} \, \mathcal{L}_t(s_t) := \mathbb{E}_{y \sim p_t} \Bigg[ \sum_{x \neq y} w_{x,y} \Big( s_t(x,y)$$

$$- \frac{p_t(x)}{p_t(y)} \log s_t(x,y) + K\left(\frac{p_t(x)}{p_t(y)}\right) \Big) \Bigg], \tag{4}$$

where the weight $w_{x,y} \geq 0$ and $K(z) = z(\log z - 1)$ serves as a normalizing function that ensures the objective remains positive. When setting $w_{x,y} = Q_t(y,x)$, Benton et al. (2024) and Chen & Ying (2024) demonstrate that $\int \mathcal{L}_t \mathrm{d}t$ closely approximates the KL-divergence between the data distribution and the distribution generated by the discrete diffusion. For the remainder of this paper, we consistently set $w_{x,y} = Q_t(y,x)$.

Given training data points $\{x^{(1)}, x^{(2)}, \ldots, x^{(n)}\}$, the empirical counterpart of $\mathcal{L}_t$ is formulated as:

$$\widehat{\mathcal{L}}_t(s_t) = \frac{1}{n} \sum_{k=1}^{n} \mathbb{E}_{y \sim p_t(\cdot|x^{(k)})} \Bigg[ \sum_{x \neq y} Q_t(y,x)$$

$$\cdot \left( s_t(x,y) - \frac{p_t(x^{(k)},x)}{p_t(x^{(k)},y)} \log s_t(x,y) \right) \Bigg]. \tag{5}$$

Here, $p_t(x,y)$ denotes the transition probability from state $x$ at time 0 to state $y$ at time $t$ under the forward process (1), which can be explicitly computed using the transition matrix $Q_t$.

## 2.2. Discrete Diffusion for Sequence Generation

The cardinality of the data support $|\mathcal{X}| = K^d$ is exponentially large, making it computationally prohibitive to explicitly define a full transition matrix and simulate the Markov process directly on $\mathcal{X}$. Consequently, it is standard practice to consider conditionally independent forward processes on each entry of $x$, which is widely adopted in large-scale

discrete diffusion models (Lou et al., 2023; Nie et al., 2025; Austin et al., 2021). Viewing each entry in $x$ as a token, practical discrete diffusion models evolve individual token independently according to a conditional absorbing or uniform transition kernel.

For a clean sequence $x_0 = (x_0^1, \ldots, x_0^d)^\top$, each token evolves independently according to a linear interpolation. The corrupted sequence $x_t$ admits a factorized conditional probability due to the independent evolution:

$$p_t(x_t|x_0) = \prod_{i=1}^{d} p_t(x_t^i|x_0^i), \quad \text{where}$$

$$p_t(x_t^i|x_0^i) = \alpha_t \mathbb{1}(x_t^i, x_0^i) + (1 - \alpha_t)p_\infty(x_t^i).$$

Here $\alpha_t \in (0,1]$ is a time-dependent rate function, $\mathbb{1}(x_t^i, x_0^i) = 1$ if $x_t^i = x_0^i$, and $\mathbb{1}(x_t^i, x_0^i) = 0$ if $x_t^i \neq x_0^i$. In particular, for uniform forward process, it holds that

$$p_t(x_t^i|x_0^i) = \alpha_t \mathbb{1}(x_t^i, x_0^i) + \frac{1 - \alpha_t}{K}. \tag{6}$$

For absorbing/mask forward process, it holds that

$$p_t(x_t^i|x_0^i) = \alpha_t \mathbb{1}(x_t^i, x_0^i) + (1 - \alpha_t) \mathbb{1}(x_t^i, \mathsf{M}), \tag{7}$$

where $\mathsf{M}$ denotes the special mask token, also referred to as the absorbing state. The conditional probability yields the marginal distribution of $x_t$:

$$p_t(x_t) = \mathbb{E}_{x_0 \sim P_{\text{data}}}[p_t(x_t|x_0)] = \sum_{x_0} p_t(x_t|x_0)p_0(x_0).$$

The following results characterize the dynamic in $p_t$ as $t$ increases. It suffices to identify the corresponding transition matrix $Q_t$ for token-wise independent forward processes.

● Uniform Transition. The transition matrix for the uniform process is given by

$$Q_t^{\text{unif}}(x,y) = \begin{cases} -d\beta_t \frac{K-1}{K} & \text{if } x = y, \\ \frac{\beta_t}{K} & \text{if } x^i \neq y^i, x^{-i} = y^{-i}, \\ 0 & \text{otherwise.} \end{cases} \tag{8}$$

Here we denote $\beta_t = -\alpha_t'/\alpha_t$. The diagonal entry of $Q_t^{\text{unif}}(x,y)$ accounts for the total rate of leaving state $y$ across all $d$ positions, where each position can transition to any of the $K - 1$ other tokens. The off-diagonal rate $\beta_t/K$ represents the uniform probability of transitioning to any specific token, reflecting the symmetry of the uniform stationary distribution and allowing bidirectional transitions

between any pair of tokens at each position.

● Absorbing Transition. The transition matrix for the absorbing process is given by

$$
Q_t^{\mathrm{absorb}}(x, y) = \begin{cases} -\beta_t \bar{M}(x) & \text{if } x = y, \\ \beta_t & \text{if condition } (\star), \\ 0 & \text{otherwise,} \end{cases} \quad (9)
$$

where condition $(\star) = \{x^i = \mathsf{M}, y^i \neq \mathsf{M}, x^{-i} = y^{-i}\}$ and $\bar{M}(x) = \sum_{i=1}^d \mathbb{1}(x \neq \mathsf{M})$. In contrast to the uniform process, the absorbing process is unidirectional: tokens can only transition from unmasked to masked states. The diagonal entry $Q_t^{\mathrm{absorb}}(y, y) = -\beta_t \bar{M}(y)$ represents the total rate at which state $y$ transitions away, which is proportional to the number of unmasked tokens. The off-diagonal entries correspond to masking transitions, where a single unmasked token at position $i$ is replaced by the masked token $\mathsf{M}$ at rate $\beta_t$. These transition matrix characterizations enable efficient computation of the backward processes and discrete scores required for sampling, leveraging their sparse structure. Interested readers may find the formal derivations in Appendix A.

## 3. Statistical Theory

In this section, we develop a theoretical framework connecting the design of forward corruption processes to generalization performances. We first establish statistical distribution estimation guarantees of discrete diffusion model with a general forward process (Section 3.1), then specialize to sequence modeling where we compare performance of different forward processes (Section 3.2).

### 3.1. Generalization Bound

We begin with a decomposition of the distribution estimation error that applies to any discrete diffusion model, regardless of the choice on the forward process.

**Proposition 3.1.** *Given any early stopping time $t_0 \geq 0$ and terminal time $T \geq 0$, let $\widehat{P}_{T-t_0}$ be the distribution generated by (3). Then it holds that*

$$
\mathrm{TV}\left(P_{\mathrm{data}}, \widehat{P}_{T-t_0}\right) \leq \underbrace{\mathrm{TV}\left(P_{\mathrm{data}}, P_{t_0}\right)}_{\text{Early stopping error}} + \underbrace{\mathrm{TV}(P_T, P_\infty)}_{\text{Mixing error}}
$$

$$
+ \underbrace{\sqrt{\frac{1}{2} \int_{t_0}^T \mathcal{L}_t(s_t) \mathrm{d}t}}_{\text{Estimation error}}.
$$

*Here $\mathcal{L}_t(s_t)$ is defined in (4) with $w_{x,y} = Q_t(y, x)$.*

The proof is provided in Appendix B.1. Proposition 3.1

demonstrates three distinct error sources for distribution estimation. First, the early stopping error measures how much the forward process has corrupted the data at stopping time $t_0$. Second, the mixing error quantifies how close the terminal distribution $P_T$ is to the stationary distribution $P_\infty$. Finally, the estimation error captures the quality of the discrete score estimator.

We focus our analysis on the estimation error, as it is the dominant term and primary bottleneck for distribution estimation. The early stopping and mixing errors can be controlled through careful selection of $t_0$ and $T$, which we instantiate concretely in Section 3.2 for the uniform and absorbing processes. We consider empirical risk minimizers of (5) with bounded range:

$$
\widehat{s}_t \in \arg\min \widehat{\mathcal{L}}_t(s_t), \quad \text{s.t.} \quad \epsilon \leq s_t(x, y) \leq C_t. \quad (10)
$$

Here $\epsilon \in (0, 1)$ is a fixed hyperparameter and $C_t > 0$ is a time-dependent hyperparameter that controls the range of the probability ratio. The constraints $\epsilon \leq s_t(x, y) \leq C_t$ ensure numerical stability of the logarithmic terms in $\mathcal{L}_t$ and regularize the empirical ratio $\bar{p}_t(x)/\bar{p}_t(y)$ against statistical noise at both extremes. These constraints are standard in discrete diffusion implementations and control the hypothesis class complexity (Lou et al., 2023; Wasserman, 2004). As demonstrated in Lemma B.1, the empirical risk minimizer admits a closed-form solution

$$
\widehat{s}_t(x, y) := \mathrm{clip}\left(\frac{\bar{p}_t(x)}{\bar{p}_t(y)}, \epsilon, C_t\right) \quad \text{for } x, y \in \mathcal{X}, \quad (11)
$$

where $\bar{p}_t$ corresponds to the diffused distribution in (1) at time $t$ with the initial distribution being the empirical distribution $\bar{p}_0(x) = \frac{1}{n} \sum_{k=1}^n \delta\{x = x^{(k)}\}$ of the dataset $\{x^{(k)}\}_{k=1}^n$. Here $\delta$ is the Dirac delta function. Next, we establish the score estimation error of $\widehat{s}_t$ for an arbitrary forward process with general transition matrix $Q_t$.

**Lemma 3.2.** *Given any $t \geq 0$, suppose for any pair $(x, y)$ such that $p_t(y)Q_t(y, x) > 0$, we have $p_t(x)/p_t(y) \in (0, C_t]$. Then for the score estimator $\widehat{s}_t$ defined in (11), its expected score entropy at time $t$ is bounded by*

$$
\mathbb{E}[\mathcal{L}_t(\widehat{s}_t)] = \widetilde{\mathcal{O}}\left(\frac{C_t}{n} \sum_{y \in \mathcal{X}} |Q_t(y, y)| \mathbb{1}(p_t(y) > 0)\right).
$$

*Here $\widetilde{\mathcal{O}}(\cdot)$ hides the logarithmic factors.*

The proof is provided in Appendix B.3. Lemma 3.2 demonstrates that the score estimation error scales linearly with two key quantities: the discrete score range $C_t$ and the support size of the noisy data distribution $p_t$. The dependence on $C_t$ reflects the difficulty of score function approxi-

mation, while the support size dependence enters through $\sum_{y \in \mathcal{X}} |Q_t(y,y)| \mathbb{1}(p_t(y) > 0)$, which counts total transition rates over states with positive probability under the forward corruption process. When $p_t$ has full support, this summation reduces to $|\mathrm{tr}(Q_t)|$ and scales at least linearly with the support size $|\mathcal{X}|$.

*Remark* 3.3. The scaling $|\mathcal{X}|/n$ in Lemma 3.2 is consistent with fundamental limits of discrete distribution estimation. The minimax rate for estimating a distribution over $\mathcal{X}$ from $n$ samples is $\Theta(\sqrt{|\mathcal{X}|/n})$ in TV distance (Han et al., 2015, Theorem 1). Since Proposition 3.1 shows the estimation error contributes $\sqrt{\int \mathcal{L}_t(\widehat{s}_t)\mathrm{d}t}$ to the TV bound, matching this minimax rate requires $\mathcal{L}_t(\widehat{s}_t) = \Omega(|\mathcal{X}|/n)$. Our bound is thus tight in its dependence on $|\mathcal{X}|$.

Lemma 3.2 reveals that forward processes simultaneously maintain small discrete score ranges $C_t$ and sparse support of $p_t$ can achieve better sample complexity. This insight motivates our analysis in Section 3.2, where we investigate how the choice of forward process affects these quantities in sequence modeling.

## 3.2. Sequence Modeling

We now specialize the general statistical framework to sequence modeling and compare the uniform and absorbing processes. We first characterize how each process determines the discrete score range $C_t$ and support evolution, the two quantities identified as critical in Lemma 3.2. Then we analyze the three error components in Proposition 3.1. To ease the presentation, we defer the proofs of lemmas to Appendix C.

We begin with deriving the discrete scores for the conditional independent uniform process.

**Lemma 3.4.** *Consider the conditional independent uniform process with transition matrix $Q_t^{\mathrm{unif}}$ given in (8). For any time $t > 0$, the discrete score $p_t(x)/p_t(y)$ such that $p_t(y)Q_t^{\mathrm{unif}}(y,x) > 0$ satisfies $x^i \neq y^i, x^{-i} = y^{-i}$ for some $i \in [d]$ and*

$$\frac{p_t(x)}{p_t(y)} = \frac{\alpha_t \mathbb{P}(X_0^i = x^i | X_t^{-i} = y^{-i}) + \frac{1-\alpha_t}{K}}{\alpha_t \mathbb{P}(X_0^i = y^i | X_t^{-i} = y^{-i}) + \frac{1-\alpha_t}{K}}.$$

*Moreover, these discrete scores are bounded by*

$$C_t^{\mathrm{unif}} = \max_{\substack{x,y:\, x^i \neq y^i, \\ x^{-i} = y^{-i}}} \frac{\alpha_t \mathbb{P}(X_0^i = x^i | X_t^{-i} = y^{-i}) + \frac{1-\alpha_t}{K}}{\alpha_t \mathbb{P}(X_0^i = y^i | X_t^{-i} = y^{-i}) + \frac{1-\alpha_t}{K}}.$$

Lemma 3.4 characterizes the discrete scores for the uniform process, where transitions occur between sequences differing in exactly one token. The score range $C_t^{\mathrm{unif}}$ critically de-

pends on the probability $\mathbb{P}(X_0^i = x^i | X_t^{-i} = y^{-i})$ that token $x^i$ appears at the $i$-th position given the noisy information $y^{-i}$ at all other positions. It captures the maximum ratio of such conditional probabilities across all token pairs and positions, weighted by the noise level $\alpha_t = \exp\left(-\int_0^t \beta_s \, \mathrm{d}s\right)$.

Next, we derive the discrete scores for the conditional independent absorbing process.

**Lemma 3.5.** *Consider the conditional independent absorbing process with transition matrix $Q_t^{\mathrm{absorb}}$ given in (9). For any time $t > 0$, the discrete score $p_t(x)/p_t(y)$ such that $p_t(y)Q_t^{\mathrm{absorb}}(y,x) > 0$ satisfies $x^i \neq \mathsf{M}, y^i = \mathsf{M}, x^{-i} = y^{-i}$ for some $i$, and*

$$\frac{p_t(x)}{p_t(y)} = \frac{\alpha_t}{1-\alpha_t} \mathbb{P}(X_0^i = x^i | X_0^A = y^A),$$

*where $A = \{j \in [d] : y^j \neq \mathsf{M}\}$. Moreover, these discrete scores are bounded by*

$$C_t^{\mathrm{absorb}} = \frac{\alpha_t}{1-\alpha_t} \max_{\substack{(y^i, y^A):i \in [d], \\ A \subseteq [d]\setminus\{i\}}} \mathbb{P}(X_0^i = y^i | X_0^A = y^A).$$

Lemma 3.5 characterizes the discrete scores for the absorbing process. It reveals that the instant transition only happens between sequences varying at exactly one position, where the source sequence has some token and the target sequence has $\mathsf{M}$. Each score ratio has the form $\frac{\alpha_t}{1-\alpha_t}\mathbb{P}(X_0^i = x^i | X_0^A = y^A)$, representing the conditional probability of an unmasked token given the observed unmasked context $y^A$, scaled by $\alpha_t/(1-\alpha_t)$.

Crucially, this time-dependent factor decays exponentially to zero as $t \to \infty$, causing $C_t^{\mathrm{absorb}}$ to vanish at late times. This is in stark contrast to $C_t^{\mathrm{unif}}$, which remains bounded away from zero. This qualitative difference in score dynamics drives the distinct sample complexity behaviors of the two processes.

Having characterized the discrete score structures, we now compare their implications for score estimation errors. To facilitate comparison, we define the cumulative score estimation complexity over a time interval $[t_0, T]$ as

$$\mathcal{I}_{[t_0,T]}(C_t, Q_t) := \int_{t_0}^T \frac{C_t}{n} \sum_{y \in \mathcal{X}} |Q_t(y,y)| \mathbb{1}(p_t(y) > 0) \, \mathrm{d}t,$$

which captures the dominant terms in Lemma 3.2. This quantity provides a tractable proxy for comparing the intrinsic statistical difficulty of different forward processes. The following lemmas 3.6 and 3.7 provide upper bounds on $\mathcal{I}_{[t_0,T]}$ for uniform and absorbing processes respectively.

**Lemma 3.6.** *Consider the conditional independent uniform*

*process with transition matrix $Q_t^{\text{unif}}$ given in* (8) *under $\beta_t \equiv \beta > 0$ and $C_t^{\text{unif}}$ defined in Lemma* 3.4. *Then for any $t_0, T > 0$ satisfying $t_0 \leq T$, we have*

$$\mathcal{I}_{[t_0, T]}^{\text{unif}}(C_t^{\text{unif}}, Q_t^{\text{unif}}) \leq \frac{dK^{d-1}(K-1)}{n}$$
$$\cdot \left( \beta(T - t_0) + K \log \frac{1 - e^{-\beta T}}{1 - e^{-\beta t_0}} \right).$$

**Lemma 3.7.** *Consider the conditional independent absorbing process with transition matrix $Q_t^{\text{absorb}}$ given in* (9) *under $\beta_t \equiv \beta > 0$ and $C_t^{\text{absorb}}$ defined in Lemma* 3.5. *Let $p_t$ be the marginal distribution generated by $Q_t^{\text{absorb}}$, and $C_t^{\text{absorb}} = \sup \left\{ \frac{p_t(x)}{p_t(y)} : p_t(y)Q_t^{\text{absorb}}(y, x) > 0 \right\}$. Then for any $t_0, T > 0$ satisfying $t_0 \leq T$, we have*

$$\mathcal{I}_{[t_0, T]}^{\text{absorb}}(C_t^{\text{absorb}}, Q_t^{\text{absorb}}) = \frac{2^{d-1} d |\mathcal{X}^{\text{data}}|}{n} \log \frac{1 - e^{-\beta T}}{1 - e^{-\beta t_0}}.$$

*Here $\mathcal{X}^{\text{data}} := \{x \in \mathcal{X} : p_{\text{data}}(x) > 0\}$ denotes the data support.*

The bounds in Lemmas 3.6 and 3.7 follow from direct computation using the score characterizations in Lemmas 3.4 and 3.5. Here $\beta_t \equiv \beta$ is assumed only to obtain closed forms, and a general schedule leaves the comparison unchanged. We note that these estimates can be conservative: the uniform bound, for instance, treats all $K^d$ states equally without exploiting concentration of probability mass. Section 4 refines these results with tighter, structure-aware bounds that leverage the concentration and sparsity-preserving properties specific to each process. Nonetheless, the present results reveal a fundamental architectural distinction:

**Dense transitions in uniform process** Lemma 3.6 establishes that $\mathcal{I}^{\text{unif}}$ scales with the full support size $K^d$. This scaling is inherent to the uniform transition matrix's dense transition structure: at each time step, every token position can transition uniformly to any of the $K$ vocabulary tokens, creating a diffuse support that expands across the entire space $\mathcal{X}$. Even when the data distribution $p_{\text{data}}$ has sparse support, the forward corruption $p_t$ still spreads mass across all $K^d$ sequences.

**Sparsity preservation in absorbing process** In contrast, Lemma 3.7 shows that $\mathcal{I}^{\text{absorb}}$ scales only with the data support size $|\mathcal{X}^{\text{data}}| \leq K^d$. This efficiency stems from the absorbing mechanism: tokens can only transition to the mask token M rather than diffusing across all vocabulary tokens. Consequently, the absorbing process preserves a support structure aligned with the data support's intrinsic sparsity.

**Implications for high-dimensional regimes** As the vocabulary size $K$ and sequence length $d$ increase, which is typical in modern language modeling, the gap between these complexity measures can be substantial. For distributions with $|\mathcal{X}^{\text{data}}| \ll K^d$ (e.g., natural language where only a tiny fraction of possible sequences form meaningful text), the uniform process's dense transitions force learning over an exponentially large space, while the absorbing process's sparsity preservation confines learning to the data-relevant support, leading to superior sample efficiency. Conversely, when the data distribution has dense support with $|\mathcal{X}^{\text{data}}| \approx K^d$, both processes scale similarly, and the advantage of absorbing processes diminishes. This formalization explains why masked diffusion has proven particularly effective for structured discrete domains like language, where intrinsic sparsity is prevalent (He et al., 2023; Ou et al., 2024; Sahoo et al., 2024).

Finally, we provide the distribution estimation guarantees for both the uniform and absorbing processes.

**Theorem 3.8.** *Let $P_t^{\text{unif}}$ and $P_t^{\text{absorb}}$ denote the distribution generated by the forward process* (1) *with the uniform transition matrix in* (8) *and absorbing transition matrix in* (9), *respectively. For every $t_0, T > 0$ and any $P_t \in \{P_t^{\text{unif}}, P_t^{\text{absorb}}\}$, we have*

$$\text{TV}(P_{\text{data}}, P_{t_0}) \leq d(1 - \alpha_{t_0}) \quad \text{and} \quad \text{TV}(P_T, P_\infty) \leq d\alpha_T.$$

*Moreover, denote $\widehat{P}_{T-t_0}^{\text{unif}}$ and $\widehat{P}_{T-t_0}^{\text{absorb}}$ as the distributions generated by the backward process* (3), *each equipped with its respective transition matrix and the score estimator* (11) *trained under the corresponding forward process. Let $\beta_t \equiv \beta > 0$, $t_0 = 1/(\beta n)$ and $T = \log n/\beta$. Then we have*

$$\text{TV}\left( P_{\text{data}}, \widehat{P}_{T-t_0}^{\text{unif}} \right) = \widetilde{\mathcal{O}}\left( \sqrt{\frac{d\beta K^{d+1}}{n}} \right),$$

*and*

$$\text{TV}\left( P_{\text{data}}, \widehat{P}_{T-t_0}^{\text{absorb}} \right) = \widetilde{\mathcal{O}}\left( \sqrt{\frac{2^d d |\mathcal{X}^{\text{data}}|}{n}} \right).$$

The proof is provided in Appendix C.5. The bounds in Theorem 3.8 confirm that the estimation error is indeed the dominant bottleneck for distribution recovery. The early stopping and mixing errors are both controlled at rate $\mathcal{O}(d/n)$ through our choice of $t_0 = 1/(\beta n)$ and $T = \log n/\beta$, which becomes negligible compared to the estimation error. The key distinction between the two processes thus emerges entirely from the score estimation complexity.

These bounds, however, are worst-case in nature. They treat

all sequences within the support as equally difficult to learn, without exploiting finer distributional structure. In Section 4, we develop refined analyses revealing distinct adaptivity mechanisms of uniform processes and absorbing processes.

## 4. Adaptivity to Distributional Structures

The preceding analysis establishes a fundamental distinction: absorbing processes scale with the data support size $|\mathcal{X}^{\text{data}}|$, whereas uniform processes scale with the full space size $|\mathcal{X}|$. We now develop structure-aware bounds that sharpen these results by exploiting additional properties of the data distribution beyond raw support size.

In real-world sequential distributions, probability mass often concentrates on a small subset of plausible sequences (Manning & Schutze, 1999; Piantadosi, 2014). For instance, natural language must obey grammar and remain semantically coherent; these constraints induce strong dependencies among tokens and render many token combinations impossible or exceedingly rare. Thus, even when the ambient space $\mathcal{X}$ is enormous, the effective distribution can exhibit a sparse structure, with most mass supported on a much smaller subset. Our refined bounds capture this sparsity differently for the two forward processes. For uniform processes, the complexity depends on how close the data distribution is to being uniform over its sparse support. For absorbing processes, the complexity can be controlled by instantaneous transition counts along the corruption path. Together, these results show that both processes can adapt to structure in the data distribution, but the absorbing process achieves such adaptivity in a more direct and natural way.

We begin with the uniform process. The empirical risk minimizer given in (11) reveals that score estimation error fundamentally depends on how accurately the empirical distribution approximates its population counterpart. This is an intrinsic statistical difficulty regardless of further parameterization choices. This approximation is reliable only when probabilities are large enough to be estimated accurately from $n$ samples. To formalize this, we define the significant sequence set as

$$\mathcal{E} := \left\{ y \in \mathcal{X} : P_{\text{data}}(X^S = y^S) \geq \frac{cd\log(2n)}{n} \text{ for} \right.$$
$$\left. \text{all } S \subseteq [d] \text{ s.t. } P(X^S = y^S) > 0 \right\}, \quad (12)$$

where $c > 0$ is some absolute constant. Note that $\mathcal{E}$ includes sequences that may have zero probability under $P_{\text{data}}$, but all their positive marginals are sufficiently concentrated. By decomposing the score estimation error into contributions from $\mathcal{E}$ and its complement, we obtain the following refined

bound.

**Lemma 4.1.** *Let* $\beta_t \equiv \beta > 0$, $t_0 = 1/(\beta n)$ *and* $T = \log n/\beta$. *Denote* $\widehat{s}^{\text{unif}}$ *as the empirical risk minimizer in* (11) *with respect to the uniform process. Then we have*

$$\int_{t_0}^{T} \mathcal{L}_t(\widehat{s}_t^{\text{unif}})\mathrm{d}t \lesssim \min\{r_0, \beta + K\}\frac{d}{np_0^{\min}}$$
$$+ (\beta + K)\mathbb{P}(\mathcal{E}^c),$$

*where* $p_0^{\min}$ *and* $p_0^{\max}$ *are the minimum and maximum of* $\mathbb{P}(X^S = y^S)$ *over* $y \in \mathcal{E}$ *and* $S \subseteq [d]$ *with* $\mathbb{P}(X^S = y^S) > 0$, *and* $r_0 := p_0^{\max}/p_0^{\min}$.

The proof of Lemma 4.1 is provided in Appendix D.1. The bound decomposes into two terms with distinct origins. The first term scaling with $r_0/(np_0^{\min})$, captures the estimation error for sequences in the significant set $\mathcal{E}$, while the second term accounts for the residual contribution from $\mathcal{E}^c$. Lemma 4.1 reveals how the uniform process adapts to the data distribution: distributions that allocate probability mass more evenly across their support admit smaller $r_0$ and larger $p_0^{\min}$, as well as concentrate most mass within $\mathcal{E}$, both of which lead to smaller estimation error. In these cases, $p_0^{\min}$ can be significantly larger than $K^{-d}$, yielding improved sample complexity than Lemma 3.6.

For the absorbing process, adaptivity arises through a different mechanism: the sparsity of valid transitions in the backward denoising process. Rather than depending on how $p_{\text{data}}$ is uniformly distributed, the absorbing process automatically exploits the structure of the data support through the masking mechanism.

**Lemma 4.2.** *Let* $\beta_t \equiv \beta > 0$, $t_0 = 1/(\beta n)$ *and* $T = \log n/\beta$. *Denote* $\widehat{s}^{\text{absorb}}$ *as the empirical risk minimizer in* (11) *under* $\epsilon = \mathcal{O}(\log n/n)$ *with respect to the absorbing process. Then we have*

$$\int_{t_0}^{T} \mathcal{L}_t(\widehat{s}_t^{\text{absorb}})\mathrm{d}t = \widetilde{\mathcal{O}}\left(\frac{\mathcal{K}^{\max}}{n}\right).$$

*Here we denote* $\mathcal{K}^{\max} = \max_{t \in [t_0, T]} \mathcal{K}_t$ *with*

$$\mathcal{K}_t := \sum_{y:\, p_t(y) > 0} p_t(y)\big|\big\{x : p_t(x) > 0,\, x^{-i} = y^{-i},$$
$$x^i \neq \mathsf{M} = y^i \text{ for some } i\big\}\big|.$$

The proof of Lemma 4.2 is provided in Appendix D.2. Here $\mathcal{K}^{\max}$ represents the maximal expected number of valid unmasking transitions from a noisy sequence $y \sim p_t$, capturing the structural sparsity of the absorbing process's instant transitions. For structured distributions like natural language, where only a small subset of token completions are valid

at each masked position, $\mathcal{K}^{\max}$ can be dramatically smaller than the vocabulary size $K$.

The two processes adapt to distributional structure in fundamentally different ways.

**Uniform processes benefit from uniformity** When $p_{\text{data}}$ is close to uniform and concentrates on a sparse set $\mathcal{E}$, we have $p_0^{\min} \approx 1/|\mathcal{S}|$ and $r_0 \approx 1$, yielding complexity $\widetilde{\mathcal{O}}(d|\mathcal{S}|/n)$. However, if $p_{\text{data}}$ is highly nonuniform, the ratio $r_0$ can be large and $p_0^{\min}$ small, degrading the bound.

**Absorbing processes benefit from sparsity** At each masked position, only tokens appearing in valid data sequences contribute to $\mathcal{K}_t$. Consider an extreme case where, at each position $i$, only $k \ll K$ tokens ever appear in the data. Then $\mathcal{K}^{\max} \leq k$, yielding complexity $\widetilde{\mathcal{O}}(k/n)$ regardless of how $p_{\text{data}}$ distributes mass among valid sequences.

**Numerical Results** To validate our theoretical findings, we conduct synthetic experiments regarding the comparative performance of absorbing and uniform forward processes under varying distributional structures. While the superiority of mask/absorbing diffusion has been extensively demonstrated in language benchmarks (He et al., 2023; Sahoo et al., 2024; Nie et al., 2025; Ye et al., 2025), synthetic experiments allow us to systematically control the key distributional parameters, such as data support size and probability uniformity, which our theory identifies as critical.

*Table 1.* Normalized $\ell_1$ distance ($\times 10^{-4}$, lower is better) for absorbing and uniform processes. Varying transition sparsity with $p_0^{\min} = 0.001$.

| Transition Sparsity | Absorbing | Uniform |
|---|---|---|
| 10 | **25.27± 1.68** | 35.38± 2.29 |
| 20 | **28.48 ± 0.61** | 36.49± 3.30 |
| 40 | **37.19± 1.02** | 39.66± 1.33 |
| 60 | 43.01± 1.84 | 43.03± 1.66 |
| 80 | 45.31± 1.34 | **45.17± 1.45** |

*Table 2.* Normalized $\ell_1$ distance ($\times 10^{-4}$) for absorbing and uniform processes. Varying transition sparsity with $p_0^{\min} = 0.004$.

| Transition Sparsity | Absorbing | Uniform |
|---|---|---|
| 10 | **22.66± 1.36** | 35.99± 3.34 |
| 20 | **28.48 ± 2.19** | 34.43± 2.22 |
| 40 | **37.29± 1.10** | 38.36± 1.86 |
| 60 | 42.27± 0.86 | **43.03± 0.91** |
| 80 | 45.61± 1.21 | **45.07± 1.75** |

We generate synthetic sequential data from homogeneous Markov chains over vocabulary size $K = 100$ and sequence length $d = 12$. The transition matrices control the transition sparsity $S \in \{10, 20, 40, 60, 80\}$, where only $S$ neighbors have significant transition probability from each token. We also vary the minimum transition probability $p_0^{\min} \in \{0.001, 0.004\}$ to control uniformity. We train transformer-based score networks following SEDD (Lou et al., 2023) and evaluate using the normalized $\ell_1$ distance $\mathcal{D}(\widehat{Q}, Q^*) = \|\widehat{Q} - Q^*\|_1/(2K)$ between estimated and true transition matrices. We provide results in Tables 1-2, with mean normalized $\ell_1$ distance and standard deviation over 10 repetitions. Full details are in Appendix E.

The results align with our theoretical analysis. (i) As shown in Tables 1-2, absorbing processes perform better when transitions are sparse, while this advantage decreases as the data support becomes denser. This observation is consistent with Lemma 4.2 that the complexity scales with $\mathcal{K}_{\max}$; (ii) uniform processes improve relatively as $S$ increases, achieving better performance at $S \geq 60$; (iii) we observe that larger values of $p_0^{\min}$ benefits uniform processes, aligning with Lemma 4.1. These findings confirm that absorbing processes exploit transition sparsity while uniform processes exploit probability uniformity—complementary mechanisms that explain masked diffusion's empirical success in language, where valid next-token sets are inherently sparse.

## 5. Conclusion

This paper develops a statistical learning theory for discrete diffusion models with an explicit focus on how the *forward corruption kernel* governs generalization. Our analysis highlights the advantage of masked/absorbing processes over uniform replacement: the absorbing process scales with the effective data support, whereas the uniform process scales with the full ambient state space. We further refine these generalization bounds by deriving structure-aware ones: uniform processes benefit from uniformity of probability mass, while absorbing processes exploit sparsity in valid token completions.

There are several promising directions for future work. For example, it would be valuable to extend the comparison to richer forward kernel families (e.g., mixtures of masking and replacement, non-uniform replacement, or learned kernels). In addition, connecting our statistical bounds more tightly to computational consideration may lead to end-to-end guarantees of discrete diffusion models.

## Impact Statement

This paper presents work whose goal is to advance the field of Machine Learning. There are many potential societal consequences of our work, none which we feel must be specifically highlighted here.

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

## A. Deferred Proofs for Section 2.2

### A.1. Deriving the transition matrix in (8)

We start with formulating the conditional ODE of a single token. Utilizing the marginal distribution of a single token in (6), for any $i \in [d]$, we have

$$\frac{\mathrm{d}p_t(x^i|x_0^i)}{\mathrm{d}t} = \alpha_t' \left( \mathbb{1}(x^i, x_0^i) - \frac{1}{K} \right) = -\frac{\alpha_t'}{\alpha_t} \left( \frac{1}{K} - p_t(x^i|x_0^i) \right).$$

The above ODE can be further written as

$$\frac{\mathrm{d}p_t(x^i|x_0^i)}{\mathrm{d}t} = \sum_{y^i \in [K]} \beta_t \left( \frac{1}{K} - \mathbb{1}(x^i, y^i) \right) p_t(y^i|x_0^i). \tag{13}$$

To derive the transition matrix $Q_t$, it suffices to show that $Q_t$ satisfies the ODE

$$\frac{\mathrm{d}p_t(x)}{\mathrm{d}t} = Q_t(x, \cdot)p_t(\cdot) = \sum_{y \in [K]^d} Q_t(x, y)p_t(y), \quad \text{with} \quad p_0 = p_{\text{data}}.$$

Note that $\frac{\mathrm{d}p_t(x)}{\mathrm{d}t}$ is the expectation of its conditional counterparts,

$$\frac{\mathrm{d}p_t(x)}{\mathrm{d}t} = \mathbb{E}_{x_0 \sim p_{\text{data}}} \frac{\mathrm{d}p_t(x|x_0)}{\mathrm{d}t}.$$

Applying the conditional independence, i.e. $p_t(x|x_0) = \prod_{i=1}^d p_t(x^i|x_0^i)$ we can write $\frac{\mathrm{d}p_t(x|x_0)}{\mathrm{d}t}$ as

$$\frac{\mathrm{d}p_t(x|x_0)}{\mathrm{d}t} = \sum_{i=1}^d \frac{\mathrm{d}p_t(x^i|x_0)}{\mathrm{d}t} \cdot \frac{p_t(x|x_0)}{p_t(x^i|x_0)} = \sum_{i=1}^d \frac{\mathrm{d}p_t(x^i|x_0)}{\mathrm{d}t} \cdot p_t(x^{-i}|x_0).$$

Plugging (13) into the above equality gives

$$\frac{\mathrm{d}p_t(x|x_0)}{\mathrm{d}t} = \sum_{i=1}^d \left[ \sum_{y^i \in [K]} \beta_t \left( \frac{1}{K} - \mathbb{1}(x^i, y^i) \right) p_t(y^i|x_0^i) \right] p_t(x^{-i}|x_0)$$

$$= \mathbb{1}(x, y)d\beta_t \left( \frac{1}{K} - 1 \right) p_t(y|x_0) + \sum_{\substack{y^i \neq x^i, \ y^{-i}=x^{-i} \\ \text{for some } i}} \frac{\beta_t}{K} p_t(y|x_0).$$

Moreover, taking expectation over $x_0$ yields

$$\frac{\mathrm{d}p_t(x)}{\mathrm{d}t} = -\mathbb{1}(x, y)d\beta_t \left( 1 - \frac{1}{K} \right) p_t(y) + \sum_{\substack{y^i \neq x^i, \ y^{-i}=x^{-i} \\ \text{for some } i}} \frac{\beta_t}{K} p_t(y) = \sum_{y \in [K]^d} Q_t(x, y)p_t(y).$$

Therefore, we can conclude that $Q_t$ generates $p_t$.

### A.2. Deriving the transition matrix in (9)

We start with formulating the conditional ODE of a single token. Utilizing the marginal distribution of a single token in (7), for any $i \in [d]$, we have

$$\frac{\mathrm{d}p_t(x^i|x_0^i)}{\mathrm{d}t} = \alpha_t' \left( \mathbb{1}(x^i, x_0^i) - \mathbb{1}(x^i, \mathsf{M}) \right) = -\frac{\alpha_t'}{\alpha_t} \left( \mathbb{1}(x^i, \mathsf{M}) - p_t(x^i|x_0^i) \right).$$

The above ODE can be further written as

$$\frac{\mathrm{d}p_t(x^i|x_0^i)}{\mathrm{d}t} = \sum_{y^i \in [K]} \beta_t \left( \mathbb{1}(x^i, \mathsf{M}) - \mathbb{1}(x^i, y^i) \right) p_t(y^i|x_0^i). \tag{14}$$

To derive the transition matrix $Q_t$, it suffices to show that $Q_t$ satisfies the ODE

$$\frac{\mathrm{d}p_t(x)}{\mathrm{d}t} = Q_t(x, \cdot)p_t(\cdot) = \sum_{y \in [K]^d} Q_t(x, y)p_t(y), \quad \text{with} \quad p_0 = p_{\mathrm{data}}.$$

Note that $\frac{\mathrm{d}p_t(x)}{\mathrm{d}t}$ is the expectation of its conditional counterparts,

$$\frac{\mathrm{d}p_t(x)}{\mathrm{d}t} = \mathbb{E}_{x_0 \sim p_{\mathrm{data}}} \frac{\mathrm{d}p_t(x|x_0)}{\mathrm{d}t}.$$

Applying the conditional independence, i.e. $p_t(x|x_0) = \prod_{i=1}^{d} p_t(x^i|x_0^i)$ we can write $\frac{\mathrm{d}p_t(x|x_0)}{\mathrm{d}t}$ as

$$\frac{\mathrm{d}p_t(x|x_0)}{\mathrm{d}t} = \sum_{i=1}^{d} \frac{\mathrm{d}p_t(x^i|x_0)}{\mathrm{d}t} \cdot \frac{p_t(x|x_0)}{p_t(x^i|x_0)} = \sum_{i=1}^{d} \frac{\mathrm{d}p_t(x^i|x_0)}{\mathrm{d}t} \cdot p_t(x^{-i}|x_0).$$

Plugging (14) into the above equality gives

$$\frac{\mathrm{d}p_t(x|x_0)}{\mathrm{d}t} = \sum_{i=1}^{d} \left[ \sum_{y^i \in [K]} \beta_t \left( \mathbb{1}(x^i, \mathsf{M}) - \mathbb{1}(x^i, y^i) \right) p_t(y^i|x_0^i) \right] p_t(x^{-i}|x_0)$$

$$= \mathbb{1}(x, y) \sum_{i=1}^{d} \beta_t (\mathbb{1}(x^i, \mathsf{M}) - 1) p_t(y|x_0) + \sum_{\substack{y^i \neq x^i, \, y^{-i} = x^{-i} \\ \text{for some } i}} \beta_t \, \mathbb{1}(x^i, \mathsf{M}) p_t(y|x_0)$$

$$= -\mathbb{1}(x, y) \beta_t \bar{M}(x) p_t(y|x_0) + \sum_{\substack{y^i \neq x^i, \, y^{-i} = x^{-i} \\ \text{for some } i}} \mathbb{1}(x^i, \mathsf{M}) \beta_t p_t(y|x_0).$$

Moreover, taking expectation over $x_0$ yields

$$\frac{\mathrm{d}p_t(x)}{\mathrm{d}t} = -\mathbb{1}(x, y) \beta_t \bar{M}(x) p_t(y) + \sum_{\substack{y^i \neq x^i, \, y^{-i} = x^{-i} \\ \text{for some } i}} \mathbb{1}(x^i, \mathsf{M}) \beta_t p_t(y) = \sum_{y \in [K]^d} Q_t(x, y) p_t(y).$$

Therefore, we can conclude that $Q_t$ generates $p_t$.

## B. Deferred Lemmas and Proofs for Section 3.1

### B.1. Proof of Proposition 3.1

Let $P_t^{\leftarrow}$ denote the marginal at time $T - t$ of the exact backward process (2) initialized at $P_T$ with the ground-truth score. Let $\widetilde{P}_t^{\leftarrow}$ denote the marginal of the same backward dynamics initialized at $P_T$ but driven by the estimated score $s_t$, and let $\widehat{P}_t^{\leftarrow}$ denote the implementable process (3) initialized at the stationary distribution $P_\infty$ with the estimated score $s_t$. Then we can decompose the TV distance into three terms.

$$\mathrm{TV}\left( P_{\mathrm{data}}, \widehat{P}_{t_0}^{\leftarrow} \right) \leq \mathrm{TV}\left( P_{\mathrm{data}}, P_{t_0}^{\leftarrow} \right) + \mathrm{TV}\left( P_{t_0}^{\leftarrow}, \widetilde{P}_{t_0}^{\leftarrow} \right) + \mathrm{TV}\left( \widetilde{P}_{t_0}^{\leftarrow}, \widehat{P}_{t_0}^{\leftarrow} \right).$$

Using the Pinsker's inequality, we have

$$\mathrm{TV}\left(P_{t_0}^{\leftarrow}, \widetilde{P}_{t_0}^{\leftarrow}\right) \leq \sqrt{\frac{1}{2}\mathrm{KL}\left(P_{t_0}^{\leftarrow}, \widetilde{P}_{t_0}^{\leftarrow}\right)} \leq \sqrt{\frac{1}{2}\int_{t_0}^{T}\mathcal{L}_t(s_t)\mathrm{d}t},$$

The last inequality applies Proposition 2 of Chen & Ying (2024) that the KL divergence between $P_{t_0}$ and $\widetilde{P}_{t_0}$ can be bounded by the accumulated ELBO. Moreover, we have

$$\mathrm{TV}\left(\widetilde{P}_{t_0}^{\leftarrow}, \widehat{P}_{t_0}^{\leftarrow}\right) = \frac{1}{2}\left\|\widetilde{P}_{t_0}^{\leftarrow} - \widehat{P}_{t_0}^{\leftarrow}\right\|_1 = \frac{1}{2}\left\|\exp\left(\int_0^{T-t_0}\widehat{Q}_\tau\mathrm{d}\tau\right)(P_T - P_\infty)\right\|_1 \leq \frac{1}{2}\|P_T - P_\infty\|_1 = \mathrm{TV}(P_T, P_\infty).$$

Combining all the inequalities concludes the proof.

### B.2. Lemma B.1 and its Proof

**Lemma B.1.** *For any given $t > 0$, $\epsilon > 0$ and $C_t > 0$, $\widehat{s}_t$ defined in (11) satisfies*

$$\widehat{\mathcal{L}}_t(\widehat{s}_t) = \min_{s_t:\epsilon \leq s_t^{x,y} \leq C_t}\widehat{\mathcal{L}}_t(s_t).$$

*Proof.* Since there is no interaction between two entries of $\widehat{s}_t$ in $\widehat{\mathcal{L}}_t$, for any $x, y \in [K]$, we have

$$\begin{aligned}
\widehat{s}_t(x,y) &= \operatorname*{argmin}_{\epsilon \leq \widehat{s}_t(x,y) \leq C_t}\frac{1}{n}\sum_{k=1}^{n}\left(P_{v^{(k)},y}(0,t)\cdot\left(s_t^{x,y} - \frac{P_{v^{(k)},x}(0,t)}{P_{v^{(k)},y}(0,t)}\log s_t^{x,y}\right)\right)\\
&= \operatorname*{argmin}_{\epsilon \leq \widehat{s}_t(x,y) \leq C_t}\frac{1}{n}\sum_{k=1}^{n}P_{v^{(k)},y}(0,t)s_t^{x,y} - \frac{1}{n}\sum_{k=1}^{n}P_{v^{(k)},x}(0,t)\log s_t^{x,y}\\
&= \mathrm{clip}\left(\frac{\sum_{k=1}^{n}P_{v^{(k)},x}(0,t)}{\sum_{k=1}^{n}P_{v^{(k)},y}(0,t)}, \epsilon, C_t\right)\\
&= \mathrm{clip}\left(\frac{\bar{p}_t(x)}{\bar{p}_t(y)}, \epsilon, C_t\right).
\end{aligned}$$

$\square$

### B.3. Proof of Lemma 3.2

By the definition of the score entropy, we have

$$\mathcal{L}_t(\widehat{s}_t) = \mathbb{E}_{y\sim p_t}\left[\sum_{x\neq y}w_{x,y}\left(\widehat{s}_t(x,y) - \frac{p_t(x)}{p_t(y)}\log\widehat{s}_t(x,y) + \frac{p_t(x)}{p_t(y)}\log\left(\frac{p_t(x)}{p_t(y)}\right) - \frac{p_t(x)}{p_t(y)}\right)\right].$$

Notably, we can formulate the score entropy as a weighted sum of Bregman divergence associated with $f(z) = z\log z$:

$$\mathcal{L}_t(\widehat{s}_t) = \mathbb{E}_{y\sim p_t}\left[\sum_{x\neq y}w_{x,y}D_f\left(\widehat{s}_t(x,y), \frac{p_t(x)}{p_t(y)}\right)\right],$$

where

$$D_f\left(\widehat{s}_t(x,y), \frac{p_t(x)}{p_t(y)}\right) = \frac{p_t(x)}{p_t(y)} \log \frac{p_t(x)}{p_t(y)} - \widehat{s}_t(x,y) \log \widehat{s}_t(x,y) - (1 + \log \widehat{s}_t(x,y))\left(\frac{p_t(x)}{p_t(y)} - \widehat{s}_t(x,y)\right)$$

$$= \frac{p_t(x)}{p_t(y)} \log \frac{p_t(x)}{p_t(y)} - \frac{p_t(x)}{p_t(y)} - \frac{p_t(x)}{p_t(y)} \log \widehat{s}_t(x,y) + \widehat{s}_t(x,y).$$

For any fixed $x, y \in [K]$ satisfying $p_t(y) > 0$, we first bound the Bregman divergence $D_f\left(\widehat{s}_t(x,y), \frac{p_t(x)}{p_t(y)}\right)$ for different cases.

**Case I:** $\min\{p_t(x), p_t(y)\} \geq \frac{27 \log(2n)}{n}$. We set $\delta_x = \sqrt{\frac{3 \log(2n)}{n p_t(x)}}$ and $\delta_y = \sqrt{\frac{3 \log(2n)}{n p_t(y)}}$ respectively. It is checked that $\delta_x, \delta_y \in (0, 1/3]$. Then we apply Lemma F.1 with $\delta_x$ and $\delta_y$, which yields for probability at least $1 - 1/n$,

$$-\sqrt{\frac{3 \log(2n)}{n p_t(x)}} \leq \frac{\bar{p}_t(x)}{p_t(x)} - 1 \leq \sqrt{\frac{3 \log(2n)}{n p_t(x)}} \quad \text{and} \quad -\sqrt{\frac{3 \log(2n)}{n p_t(y)}} \leq \frac{\bar{p}_t(y)}{p_t(y)} - 1 \leq \sqrt{\frac{3 \log(2n)}{n p_t(y)}}. \tag{15}$$

This further gives the lower bound on $\bar{p}_t(x)/\bar{p}_t(y)$:

$$\frac{\bar{p}_t(x)}{\bar{p}_t(y)} \geq \frac{p_t(x)\left(1 - \sqrt{\frac{3 \log(2n)}{n p_t(x)}}\right)}{p_t(y)\left(1 + \sqrt{\frac{3 \log(2n)}{n p_t(y)}}\right)} \geq \frac{p_t(x)\left(1 - \frac{1}{3}\right)}{p_t(y)\left(1 + \frac{1}{3}\right)} = \frac{p_t(x)}{2 p_t(y)}.$$

Recall $\widehat{s}_t(x,y) = \mathrm{clip}\left(\frac{\bar{p}_t(x)}{\bar{p}_t(y)}, \epsilon, C_t\right)$. We note that $\widehat{s}_t(x,y) \geq p_t(x)/(2 p_t(y))$. Therefore, we can control the Bregman divergence using its bounded second derivative.

$$D_f\left(\widehat{s}_t(x,y), \frac{p_t(x)}{p_t(y)}\right) \leq \frac{1}{2}\left(\frac{p_t(x)}{2 p_t(y)}\right)^{-1}\left(\widehat{s}_t(x,y) - \frac{p_t(x)}{p_t(y)}\right)^2 \leq \frac{p_t(y)}{p_t(x)}\left(\frac{\bar{p}_t(x)}{\bar{p}_t(y)} - \frac{p_t(x)}{p_t(y)}\right)^2$$

Moreover, we apply (15) to derive

$$\left(\frac{\bar{p}_t(x)}{\bar{p}_t(y)} - \frac{p_t(x)}{p_t(y)}\right)^2 \leq \left(\frac{p_t(x) + \sqrt{\frac{3 \log(2n)}{n} p_t(x)}}{p_t(y) - \sqrt{\frac{3 \log(2n)}{n} p_t(y)}} - \frac{p_t(x)}{p_t(y)}\right)^2 \leq \frac{27 \log(2n) p_t(x)}{2n (p_t(y))^2}\left(1 + \frac{p_t(x)}{p_t(y)}\right).$$

Therefore, given $p_t(x)/p_t(y) \leq C_t$, we have

$$D_f\left(\widehat{s}_t(x,y), \frac{p_t(x)}{p_t(y)}\right) \leq \frac{27 \log(2n)}{2n p_t(y)}\left(1 + \frac{p_t(x)}{p_t(y)}\right) \leq \frac{27(C_t + 1) \log(2n)}{2n p_t(y)}. \tag{16}$$

**Case II:** $p_t(y) < \frac{27 \log(2n)}{n}$. When $p_t(y)$ is sufficiently small, it suffices to bound each term in the score entropy by their range. Since $\epsilon \leq \widehat{s}_t(x,y) \leq C_t$ and $p_t(x)/p_t(y) \leq C_t$, we have

$$D_f\left(\widehat{s}_t(x,y), \frac{p_t(x)}{p_t(y)}\right) = -\frac{p_t(x)}{p_t(y)} \log \widehat{s}_t(x,y) + \frac{p_t(x)}{p_t(y)} \log\left(\frac{p_t(x)}{p_t(y)}\right) + \left(\widehat{s}_t(x,y) - \frac{p_t(x)}{p_t(y)}\right)$$

$$\leq C_t \log(1/\epsilon) + C_t \log C_t + C_t. \tag{17}$$

**Case III:** $p_t(x) < \frac{27 \log(2n)}{n}$ **and** $p_t(y) \geq \frac{27 \log(2n)}{n}$. It remains to analyze the situation where $p_t(x)$ is sufficiently small while $p_t(y)$ is significant. In this case, we decompose the Bregman divergence $D_f\left(\widehat{s}_t(x,y), \frac{p_t(x)}{p_t(y)}\right)$ into two components

and adopt a more detailed analysis.

$$D_f\left(\widehat{s}_t(x,y), \frac{p_t(x)}{p_t(y)}\right) = \underbrace{-\frac{p_t(x)}{p_t(y)}\log \widehat{s}_t(x,y) + \frac{p_t(x)}{p_t(y)}\log\left(\frac{p_t(x)}{p_t(y)}\right)}_{(i)} + \underbrace{\left(\widehat{s}_t(x,y) - \frac{p_t(x)}{p_t(y)}\right)}_{(ii)}.$$

First, we control $(i)$ by the range of $\widehat{s}_t(x,y)$ and $p_t(x)/p_t(y)$, which gives

$$(i) \le \frac{p_t(x)}{p_t(y)}\log(1/\epsilon).$$

Next, we apply the concentration bound on $\bar{p}_t(y)$ given in (15) to $(ii)$. Given $p_t(y) \ge \frac{27\log(2n)}{n}$, it holds with probability at least $1 - 1/n$ that

$$(ii) \le \frac{\bar{p}_t(x)}{\bar{p}_t(y)} - \frac{p_t(x)}{p_t(y)} \le \frac{\bar{p}_t(x)}{p_t(y)\left(1 - \sqrt{\frac{3\log(2n)}{np_t(y)}}\right)} - \frac{p_t(x)}{p_t(y)} \le \frac{3\bar{p}_t(x)}{2p_t(y)} - \frac{p_t(x)}{p_t(y)}.$$

Then we apply Lemma F.2 to bound $\bar{p}_t(x)$:

$$(ii) \le \frac{3}{2p_t(y)}\left(\sqrt{\frac{4p_t(x)\log(2n)}{n}} + \frac{4\log(2n)}{3n}\right) - \frac{p_t(x)}{p_t(y)},$$

which holds with probability at least $1 - 1/n$. Combining the upper bound of $(i)$ and $(ii)$, we obtain

$$D_f\left(\widehat{s}_t(x,y), \frac{p_t(x)}{p_t(y)}\right) \le \frac{p_t(x)}{p_t(y)}\log(1/\epsilon) + \frac{3}{2p_t(y)}\left(\sqrt{\frac{4p_t(x)\log(2n)}{n}} + \frac{4\log(2n)}{3n}\right) - \frac{p_t(x)}{p_t(y)}$$

$$\le \frac{\log(2n)\left(\log(1/\epsilon) + 12\right)}{np_t(y)}. \tag{18}$$

Finally, we combine all the cases to derive the upper bound on $\mathcal{L}_t(\widehat{s}_t)$. The score entropy $\mathcal{L}_t(\widehat{s}_t)$ can be rewritten as

$$\mathcal{L}_t(\widehat{s}_t) = \sum_{y\in\mathcal{X}} p_t(y)\,\mathbb{1}\left(\min\{p_t(x), p_t(y)\} > \frac{27\log(2n)}{n}\right)\left[\sum_{x\ne y} w_{x,y} D_f\left(\widehat{s}_t(x,y), \frac{p_t(x)}{p_t(y)}\right)\right]$$

$$+ \sum_{y\in\mathcal{X}} p_t(y)\,\mathbb{1}\left(p_t(y) < \frac{27\log(2n)}{n}\right)\left[\sum_{x\ne y} w_{x,y}\frac{p_t(x)}{p_t(y)} D_f\left(\widehat{s}_t(x,y), \frac{p_t(x)}{p_t(y)}\right)\right]$$

$$+ \sum_{y\in\mathcal{X}} p_t(y)\,\mathbb{1}\left(p_t(x) < \frac{27\log(2n)}{n}, p_t(y) \ge \frac{27\log(2n)}{n}\right)\left[\sum_{x\ne y} w_{x,y}\frac{p_t(x)}{p_t(y)} D_f\left(\widehat{s}_t(x,y), \frac{p_t(x)}{p_t(y)}\right)\right].$$

Combining the three cases, we bound each state pair's Bregman contribution by its case-specific estimate (16)-(18) and sum

over all $x, y$ with $p_t(y) > 0$.

$$\mathbb{E}[\mathcal{L}_t(\widehat{s}_t)] \leq \frac{1}{n} \sum_{y \in \mathcal{X}} p_t(y) \left[ \sum_{x \neq y} w_{x,y} \left( C_t \log(1/\epsilon) + C_t \log C_t + C_t \right) \right] + \sum_{y \in \mathcal{X}} p_t(y) \left[ \sum_{x \neq y} w_{x,y} \frac{27(C_t + 1) \log(2n)}{2n p_t(y)} \right]$$

$$+ \sum_{y \in \mathcal{X}} p_t(y) \mathbb{1} \left( p_t(y) < \frac{27 \log(2n)}{n} \right) \left[ \sum_{x \neq y} w_{x,y} \left( C_t \log(1/\epsilon) + C_t \log C_t + C_t \right) \right]$$

$$+ \sum_{y \in \mathcal{X}} p_t(y) \left[ \sum_{x \neq y} w_{x,y} \frac{\log(2n) \left( \log(1/\epsilon) + 12 \right)}{n p_t(y)} \right].$$

Simplifying the above inequality yields

$$\mathbb{E}[\mathcal{L}_t(\widehat{s}_t)] \leq \frac{\sum_{x \neq y} w_{x,y} \, \mathbb{1}(p_t(y) > 0)}{n} \left( (C_t + 28 \log(2n)) \log \frac{C_t}{\epsilon} + C_t(41 \log(2n) + 1) + 26 \log(2n) \right).$$

Taking $w_{x,y} = Q_t(y, x)$, we will have $\sum_{x \neq y} w_{x,y} \, \mathbb{1}(p_t(y) > 0) = \sum_{y \in \mathcal{X}} |Q_t(y, y)| \, \mathbb{1}(p_t(y) > 0)$. Applying it to the above inequality concludes the proof.

## C. Deferred Lemmas and Proofs for Section 3.2

### C.1. Proof of Lemma 3.4

Fix any $x, y \in [K]^d$ such that $x^i \neq y^i$ and $x^{-i} = y^{-i}$ for some $i \in [d]$. Then we can write the ratio $p_t(x)/p_t(y)$ as

$$\frac{p_t(x)}{p_t(y)} = \frac{\sum_{x_0} p_t(x|x_0) p_0(x_0)}{\sum_{x_0} p_t(y|x_0) p_0(x_0)} = \frac{\sum_{x_0} \mathbb{P}(X_t^i = x^i, X_t^{-i} = y^{-i}|X_0 = x_0) p_0(x_0)}{\sum_{x_0} \mathbb{P}(X_t^i = y^i, X_t^{-i} = y^{-i}|X_0 = x_0) p_0(x_0)}.$$

Utilizing the conditional independence and (6), we have

$$\mathbb{P}(X_t^i = x^i, X_t^{-i} = y^{-i}|X_0 = x_0) = \left( \alpha_t \mathbb{1}(x^i = x_0^i) + \frac{1 - \alpha_t}{K} \right) \mathbb{P}(X_t^{-i} = y^{-i}|X_0^{-i} = x_0^{-i}), \tag{19}$$

and similarly,

$$\mathbb{P}(X_t^i = y^i, X_t^{-i} = y^{-i}|X_0 = x_0) = \left( \alpha_t \mathbb{1}(y^i = x_0^i) + \frac{1 - \alpha_t}{K} \right) \mathbb{P}(X_t^{-i} = y^{-i}|X_0^{-i} = x_0^{-i}). \tag{20}$$

Plugging (19) and (20) into the ratio $p_t(x)/p_t(y)$ yields

$$\frac{p_t(x)}{p_t(y)} = \frac{\sum_{x_0} \left( \alpha_t \mathbb{1}(x^i = x_0^i) + \frac{1 - \alpha_t}{K} \right) \mathbb{P}(X_t^{-i} = y^{-i}|X_0^{-i} = x_0^{-i}) p_0(x_0)}{\sum_{x_0} \left( \alpha_t \mathbb{1}(y^i = x_0^i) + \frac{1 - \alpha_t}{K} \right) \mathbb{P}(X_t^{-i} = y^{-i}|X_0^{-i} = x_0^{-i}) p_0(x_0)}$$

$$= \frac{\alpha_t \mathbb{P}(X_0^i = x^i, X_t^{-i} = y^{-i}) + \frac{1 - \alpha_t}{K} \mathbb{P}(X_t^{-i} = y^{-i})}{\alpha_t \mathbb{P}(X_0^i = y^i, X_t^{-i} = y^{-i}) + \frac{1 - \alpha_t}{K} \mathbb{P}(X_t^{-i} = y^{-i})}.$$

Dividing both the numerator and denominator by $\mathbb{P}(X_t^{-i} = y^{-i})$, we derive

$$\frac{p_t(x)}{p_t(y)} = \frac{\alpha_t \mathbb{P}(X_0^i = x^i|X_t^{-i} = y^{-i}) + \frac{1 - \alpha_t}{K}}{\alpha_t \mathbb{P}(X_0^i = y^i|X_t^{-i} = y^{-i}) + \frac{1 - \alpha_t}{K}}.$$

Due to the symmetry of the ratios, the discrete score range is obtained by taking maximum over $p_t(x)/p_t(y)$ for all the pairs $(x, y)$ satisfying $x^i \neq y^i$ and $x^{-i} = y^{-i}$ for some $i$.

## C.2. Proof of Lemma 3.5

Fix any $x, y \in [K]^d$ such that $x^i \neq \mathsf{M}, y^i = \mathsf{M}$, and $x^{-i} = y^{-i}$ for some $i \in [d]$. Let $A = \{j \in [d] : y^j \neq \mathsf{M}\}$ and $\bar{A} = \{j \neq i : y^j = \mathsf{M}\}$. Then we can formulate the ratio $p_t(x)/p_t(y)$ as

$$\frac{p_t(x)}{p_t(y)} = \frac{\sum_{x_0} p_t(x|x_0)p_0(x_0)}{\sum_{x_0} p_t(y|x_0)p_0(x_0)} = \frac{\sum_{x_0} \mathbb{P}(X_t^i = x^i, X_t^A = y^A, X_t^{\bar{A}} = \mathsf{M}|X_0 = x_0)p_0(x_0)}{\sum_{x_0} \mathbb{P}(X_t^i = \mathsf{M}, X_t^A = y^A, X_t^{\bar{A}} = \mathsf{M}|X_0 = x_0)p_0(x_0)}.$$

According to the absorbing process with conditional probability in (7), the $j$-th token $x_0^j$ either turns into the absorbing state $\mathsf{M}$ or stays the same. This yields

$$\mathbb{P}(X_t^i = x^i, X_t^A = y^A, X_t^{\bar{A}} = \mathsf{M}|X_0 = x_0) = \mathbb{1}(x_0^i = x^i, x_0^A = y^A)\mathbb{P}(X_t^i = y^i, X_t^A = y^A, X_t^{\bar{A}} = \mathsf{M}|X_0 = x_0).$$

Likewise, we have

$$\mathbb{P}(X_t^i = \mathsf{M}, X_t^A = y^A, X_t^{\bar{A}} = \mathsf{M}|X_0 = x_0) = \mathbb{1}(x_0^A = y^A)\mathbb{P}(X_t^i = \mathsf{M}, X_t^A = y^A, X_t^{\bar{A}} = \mathsf{M}|X_0 = x_0).$$

Therefore, we can derive

$$\frac{p_t(x)}{p_t(y)} = \frac{\sum_{x_0} \mathbb{1}(x_0^i = x^i, x_0^A = y^A)\mathbb{P}(X_t^i = x^i, X_t^A = y^A, X_t^{\bar{A}} = \mathsf{M}|X_0 = x_0)p_0(x_0)}{\sum_{x_0} \mathbb{1}(x_0^A = y^A)\mathbb{P}(X_t^i = \mathsf{M}, X_t^A = y^A, X_t^{\bar{A}} = \mathsf{M}|X_0 = x_0)p_0(x_0)}. \tag{21}$$

Utilizing the conditional independence and (7), we have

$$\begin{aligned}
&\mathbb{P}(X_t^i = x^i, X_t^A = y^A, X_t^{\bar{A}} = \mathsf{M}|X_0^i = x^i, X_0^A = y^A, X_0^{\bar{A}} = x_0^{\bar{A}}) \\
&= \mathbb{P}(X_t^i = x^i|X_0^i = x_0^i) \cdot \prod_{j \in A} \mathbb{P}(X_t^j = y^j|X_0^j = y^j) \cdot \prod_{j \in \bar{A}} \mathbb{P}(X_t^j = \mathsf{M}|X_0^j = x_0^j) \\
&= \alpha_t^{|A|+1}(1 - \alpha_t)^{|\bar{A}|}.
\end{aligned} \tag{22}$$

Similarly, we can derive

$$\begin{aligned}
&\mathbb{P}(X_t^i = \mathsf{M}, X_t^A = y^A, X_t^{\bar{A}} = \mathsf{M}|X_0^i = x_0^i, X_0^A = y^A, X_0^{\bar{A}} = x_0^{\bar{A}}) \\
&= \mathbb{P}(X_t^i = \mathsf{M}|X_0^i = x_0^i) \cdot \prod_{j \in A} \mathbb{P}(X_t^j = y^j|X_0^j = y^j) \cdot \prod_{j \in \bar{A}} \mathbb{P}(X_t^j = \mathsf{M}|X_0^j = x_0^j) \\
&= \alpha_t^{|A|}(1 - \alpha_t)^{|\bar{A}|+1}.
\end{aligned} \tag{23}$$

Plugging (22) and (23) into (21) yields

$$\frac{p_t(x)}{p_t(y)} = \frac{\sum_{x_0} \mathbb{1}(x_0^i = x^i, x_0^A = y^A)\alpha_t^{|A|+1}(1 - \alpha_t)^{|\bar{A}|}p_0(x_0)}{\sum_{x_0} \mathbb{1}(x_0^A = y^A)\alpha_t^{|A|}(1 - \alpha_t)^{|\bar{A}|+1}p_0(x_0)} = \frac{\alpha_t}{1 - \alpha_t} \cdot \frac{\mathbb{P}(X_0^i = x^i, X_0^A = y^A)}{\mathbb{P}(X_0^A = y^A)}.$$

Noting $\mathbb{P}(X_0^i = x^i|X_0^A = y^A) = \mathbb{P}(X_0^i = x^i, X_0^A = y^A)/\mathbb{P}(X_0^A = y^A)$, we can get

$$\frac{p_t(x)}{p_t(y)} = \frac{\alpha_t}{1 - \alpha_t}\mathbb{P}(X_0^i = x^i|X_0^A = y^A).$$

For any sequence pair $(x, y)$, the corresponding transition matrix $Q_t(x, y)$ of the conditional independent absorbing process (9) between $(x, y)$ is nonzero if and only if $x = y$ or $(x, y)$ has the form $x = (\mathsf{M}, y^A, \mathsf{M}, \ldots, \mathsf{M})$ and $y = (y^i, y^A, \mathsf{M}, \ldots, \mathsf{M})$. Finally, we can obtain the maximal discrete score at time $t > 0$ by taking maximum over all the choices of $(y^i, y^A)$.

## C.3. Proof of Lemma 3.6

We aim to derive an upper bound for

$$\mathcal{I}_{[t_0,T]}^{\mathrm{unif}}(C_t^{\mathrm{unif}}, Q_t^{\mathrm{unif}}) = \int_{t_0}^{T} \frac{C_t^{\mathrm{unif}}}{n} \sum_{y \in \mathcal{X}} |Q_t^{\mathrm{unif}}(y,y)| \, \mathbb{1}(p_t^{\mathrm{unif}}(y) > 0) \, \mathrm{d}t.$$

where $p_t^{\mathrm{unif}}$ is the marginal distribution generated by $Q_t^{\mathrm{unif}}$. We first apply Lemma 3.4 to bound the score range $C_t^{\mathrm{unif}}$ by

$$C_t^{\mathrm{unif}} = 1 + \max_{\substack{x,y:\, x^i \neq y^i, \\ x^{-i} = y^{-i}}} \frac{\alpha_t \mathbb{P}(X_0^i = x^i | X_t^{-i} = y^{-i}) - \alpha_t \mathbb{P}(X_0^i = y^i | X_t^{-i} = y^{-i})}{\alpha_t \mathbb{P}(X_0^i = y^i | X_t^{-i} = y^{-i}) + \frac{1 - \alpha_t}{K}} \leq 1 + \frac{K\alpha_t}{1 - \alpha_t}.$$

Next, we analyze the term $\sum_{y \in \mathcal{X}} |Q_t^{\mathrm{unif}}(y,y)| \, \mathbb{1}(p_t^{\mathrm{unif}}(y) > 0)$, which sums the absolute values of diagonal entries of the transition matrix $Q_t^{\mathrm{unif}}$ over all states with positive probability under $p_t$. For the uniform process, $p_t^{\mathrm{unif}}$ maintains full support over $\mathcal{X}$ for all $t > 0$ according to (6). Utilizing the transition matrix $Q_t^{\mathrm{unif}}$ presented in (8) yields

$$\sum_{y \in \mathcal{X}} |Q_t^{\mathrm{unif}}(y,y)| \, \mathbb{1}(p_t^{\mathrm{unif}}(y) > 0) = dK^{d-1}(K-1)\beta, \quad \text{for all } t > 0.$$

Note that we fix $\beta_t = \beta$ and thus $\alpha_t = \exp(-\beta t)$. Finally, it follows by elementary computation that

$$\begin{aligned}
\mathcal{I}_{[t_0,T]}^{\mathrm{unif}}(C_t^{\mathrm{unif}}, Q_t^{\mathrm{unif}}) &\leq \int_{t_0}^{T} \frac{dK^{d-1}(K-1)}{n} \beta \left( 1 + \frac{K\alpha_t}{1 - \alpha_t} \right) \mathrm{d}t \\
&= \frac{dK^{d-1}(K-1)}{n} \left( \beta(T - t_0) + K \log \left( \frac{1 - e^{-\beta T}}{1 - e^{-\beta t_0}} \right) \right).
\end{aligned}$$

## C.4. Proof of Lemma 3.7

We aim to bound the cumulative score estimation complexity for the absorbing process:

$$\mathcal{I}_{[t_0,T]}^{\mathrm{absorb}}(C_t^{\mathrm{absorb}}, Q_t^{\mathrm{absorb}}) = \int_{t_0}^{T} \frac{C_t^{\mathrm{absorb}}}{n} \sum_{y \in \mathcal{X}} |Q_t^{\mathrm{absorb}}(y,y)| \, \mathbb{1}(p_t^{\mathrm{absorb}}(y) > 0) \, \mathrm{d}t.$$

where $p_t^{\mathrm{absorb}}$ is the marginal distribution generated by $Q_t^{\mathrm{absorb}}$. By Lemma 3.5, the score range $C_t^{\mathrm{absorb}}$ is naturally bounded by

$$C_t^{\mathrm{absorb}} \leq \frac{\alpha_t}{1 - \alpha_t}.$$

For the absorbing process, the support structure is fundamentally different: $p_t^{\mathrm{absorb}}$ preserves the sparsity of the data distribution. Any noisy observation $y$ with $p_t^{\mathrm{absorb}}(y) > 0$ must be a (partially) masked version of some sequence in the data support $\mathcal{X}^{\mathrm{data}}$. This observation together with the transition matrix $Q_t^{\mathrm{absorb}}$ in (9) yields

$$\sum_{y \in \mathcal{X}} |Q_t^{\mathrm{absorb}}(y,y)| \, \mathbb{1}(p_t^{\mathrm{absorb}}(y) > 0) = \sum_{x \in \mathcal{X}^{\mathrm{data}}} \sum_{k=0}^{d} \beta \binom{d}{k} k = 2^{d-1} d |\mathcal{X}^{\mathrm{data}}| \beta, \quad \text{for all } t > 0.$$

Then by direct calculation, we have

$$
\begin{aligned}
\mathcal{I}^{\text{absorb}}_{[t_0,T]}(C^{\text{absorb}}_t, Q^{\text{absorb}}_t) &= \int_{t_0}^{T} \frac{2^{d-1}d|\mathcal{X}^{\text{data}}|}{n} \cdot \frac{\beta \alpha_t}{1 - \alpha_t} \, \mathrm{d}t \\
&= \frac{2^{d-1}d|\mathcal{X}^{\text{data}}|}{n} \log \frac{1 - e^{-\beta T}}{1 - e^{-\beta t_0}}.
\end{aligned}
$$

### C.5. Proof of Theorem 3.8

We start with analyzing the early stopping error $\text{TV}(P_{\text{data}}, P_{t_0})$. Note that

$$
\text{TV}(P_{t_0}, P_{\text{data}}) = \sup_A (P_{t_0}(A) - P_{\text{data}}(A)) \leq \mathbb{P}(X_{t_0} \neq X_0).
$$

This is because for any coupling $(Z_1, Z_2)$ with $Z_1 \sim \mu$, $Z_2 \sim \nu$ and any event $A$, one has

$$
\mu(A) - \nu(A) = \mathbb{E}\big[\mathbf{1}\{Z_1 \in A\} - \mathbf{1}\{Z_2 \in A\}\big] \leq \mathbb{E}\big[\mathbf{1}\{Z_1 \neq Z_2\}\big] = \mathbb{P}(Z_1 \neq Z_2).
$$

By union bound, we have

$$
\mathbb{P}(X_t \neq X_0) \leq \sum_{i=1}^{d} \mathbb{P}(X_t^i \neq X_0^i).
$$

For either uniform process (6) and absorbing process (7), we have

$$
\mathbb{P}(X_t^i = X_0^i \mid X_0^i) \geq \alpha_t.
$$

Hence $\mathbb{P}(X_t^i \neq X_0^i) \leq 1 - \alpha_t$ for each $i \in [d]$, and therefore

$$
\mathbb{P}(X_t^i \neq X_0^i \mid X_0^i) \leq d(1 - \alpha_t).
$$

We thereby prove the upper bound on the early stopping error

$$
\text{TV}(P_{t_0}, P_{\text{data}}) \leq \mathbb{P}(X_{t_0} \neq X_0) \leq d(1 - \alpha_t).
$$

Next, we analyze the mixing error $\text{TV}(P_T, P_\infty)$. Similarly, we have

$$
\text{TV}(P_T, P_\infty) \leq \mathbb{P}(X_T \neq X_\infty) \leq \sum_{i=1}^{d} \mathbb{P}(X_T^i \neq X_\infty^i).
$$

According to the marginal distribution of the uniform process (6) and that of absorbing process (7), we can derive

$$
\mathbb{P}(X_T^i \neq X_\infty^i) \leq \alpha_T.
$$

Therefore, we obtain

$$
\text{TV}(P_T, P_\infty) \leq d\alpha_T.
$$

Now we establish the distribution estimation errors for both the uniform estimator $\widehat{P}^{\text{unif}}_{T-t_0}$ and the absorbing estimator $\widehat{P}^{\text{absorb}}_{T-t_0}$. Let $\beta_t = \beta$ and thus $\alpha_t = \exp(-\beta t)$. Applying the error decomposition in Proposition 3.1 and the score estimation

errors in Lemmas 3.6-3.7, we have

$$
\mathrm{TV}\left(P_{\mathrm{data}}, \widehat{P}_{T-t_0}^{\mathrm{unif}}\right) \leq \mathrm{TV}\left(P_{\mathrm{data}}, P_{t_0}^{\mathrm{unif}}\right) + \mathrm{TV}(P_T^{\mathrm{unif}}, P_\infty) + \sqrt{\frac{1}{2}\int_{t_0}^{T}\mathcal{L}_t(\widehat{s}_t^{\mathrm{unif}})\mathrm{d}t}
$$

$$
\lesssim d(1 - e^{-\beta t_0}) + de^{-\beta T} + \sqrt{\frac{dK^{d-1}(K-1)}{2n}\left(\beta(T-t_0) + K\log\frac{1-e^{-\beta T}}{1-e^{-\beta t_0}}\right)}.
$$

and

$$
\mathrm{TV}\left(P_{\mathrm{data}}, \widehat{P}_{T-t_0}^{\mathrm{absorb}}\right) \leq \mathrm{TV}\left(P_{\mathrm{data}}, P_{t_0}^{\mathrm{absorb}}\right) + \mathrm{TV}(P_T^{\mathrm{absorb}}, P_\infty) + \sqrt{\frac{1}{2}\int_{t_0}^{T}\mathcal{L}_t(\widehat{s}_t^{\mathrm{absorb}})\mathrm{d}t}
$$

$$
\lesssim d(1 - e^{-\beta t_0}) + de^{-\beta T} + \sqrt{\frac{2^{d-2}d|\mathcal{X}^{\mathrm{data}}|}{n}\log\frac{1-e^{-\beta T}}{1-e^{-\beta t_0}}}.
$$

Taking $t_0 = 1/(\beta n)$ and $T = \log n/\beta$ yields

$$
\mathrm{TV}\left(P_{\mathrm{data}}, \widehat{P}_{T-t_0}^{\mathrm{unif}}\right) = \widetilde{\mathcal{O}}\left(\sqrt{\frac{d\beta K^{d+1}}{n}}\right) \quad \text{and} \quad \mathrm{TV}\left(P_{\mathrm{data}}, \widehat{P}_{T-t_0}^{\mathrm{absorb}}\right) = \widetilde{\mathcal{O}}\left(\sqrt{\frac{2^d d|\mathcal{X}^{\mathrm{data}}|}{n}}\right).
$$

The proof is complete.

## D. Deferred Proofs for Section 4

### D.1. Proof of Lemma 4.1

Consider the distribution $\bar{P}_t$, which is generated by the forward process (1) with the uniform transition matrix (Lemma 3.4) and initialized at the empirical distribution $\bar{P}_0$ with probability mass $\bar{p}_0(x) = \frac{1}{n}\sum_{k=1}^{n}\mathbb{1}\{x = x^{(k)}\}$ of the dataset $\{x^{(k)}\}_{k=1}^{n}$.

As shown in Lemma D.1, the conditional probability $\mathbb{P}(X_0^i = x^i | X_t^{-i} = y^{-i})$ admits a closed-form expression as a weighted sum over the data distribution $P_{\mathrm{data}}$. This characterization also applies to discrete scores estimated from the empirical data distribution. For $x, y \in \mathcal{X}$ satisfying $x^i \neq y^i, x^{-i} = y^{-i}$ for some $i \in [d]$, we have

$$
\frac{\bar{p}_t(x)}{\bar{p}_t(y)} = \frac{\alpha_t\widehat{\mathbb{P}}(X_0^i = x^i | X_t^{-i} = y^{-i}) + \frac{1-\alpha_t}{K}}{\alpha_t\widehat{\mathbb{P}}(X_0^i = y^i | X_t^{-i} = y^{-i}) + \frac{1-\alpha_t}{K}},
$$

where $\widehat{\mathbb{P}}$ denotes the probability measure induced by sampling $X_0$ from the empirical distribution $\bar{P}_0$. Recall that in Appendix B.3, we can formulate the score estimation error as the weighted Bregman divergence associated with $f(z) = z\log z$:

$$
\mathcal{L}_t(\widehat{s}_t) = \mathbb{E}_{y \sim p_t}\left[\sum_{x \neq y} w_{x,y}D_f\left(\widehat{s}_t(x,y), \frac{p_t(x)}{p_t(y)}\right)\right].
$$

The key to upper bound $\mathbb{E}[\mathcal{L}_t(\widehat{s}_t)]$ is the concentration property of $\bar{p}_t(x)/\bar{p}_t(y)$ around $p_t(x)/p_t(y)$. When $t > t_0$ with $t_0 = 1/(\beta n)$, e.g. $\alpha_{t_0} = \mathcal{O}(1/n)$, $p_t(x)$ is bounded away from zero for any $x$ according to (6). We can show that there exists an absolute constant $c_r \geq 1$ that

$$
c_r^{-1}\frac{p_t(x)}{p_t(y)} \leq \frac{\bar{p}_t(x)}{\bar{p}_t(y)} \leq c_r\frac{p_t(x)}{p_t(y)}.
$$

For $y \in \mathcal{E}$, denote $p_0^{\min} := \min\{P(X^S = y^S) : P(X^S = y^S) \neq 0, S \subseteq [d], y \in \mathcal{E}\}$, where we take

$$\mathcal{E} := \{y \in \mathcal{X} : P(X^S = y^S) \geq 12d \log(2n)/n \text{ for any } S \subseteq [d] \text{ satisfying } P(X^S = y^S) > 0\}.$$

We apply the concentration result in Lemma D.3 to get

$$
\left| \frac{\bar{p}_t(x)}{\bar{p}_t(y)} - \frac{p_t(x)}{p_t(y)} \right| = \left| \frac{1}{\bar{p}_t(y)} (\bar{p}_t(x) - p_t(x)) + \frac{p_t(x)}{p_t(y)\bar{p}_t(y)} (\bar{p}_t(y) - p_t(y)) \right|
$$
$$
\leq \frac{c_r \alpha_t \mathbb{P}(X_0^i = x^i | X_t^{-i} = y^{-i})}{\alpha_t \mathbb{P}(X_0^i = y^i | X_t^{-i} = y^{-i}) + \frac{1-\alpha_t}{K}} \sqrt{\frac{c(d + \log n)}{n p_0^{\min}}}
$$
$$
+ \frac{c_r \alpha_t \mathbb{P}(X_0^i = y^i | X_t^{-i} = y^{-i}) \cdot p_t(x)/p_t(y)}{\alpha_t \mathbb{P}(X_0^i = y^i | X_t^{-i} = y^{-i}) + \frac{1-\alpha_t}{K}} \sqrt{\frac{c(d + \log n)}{n p_0^{\min}}},
$$

which holds for probability at least $1 - 1/n$. On the other hand, we utilize the natural score range in (17) to bound the Bregman divergence. Therefore, we can bound the score estimation error by

$$
\mathcal{L}_t(\widehat{s}_t) \leq \mathbb{E}_{y \sim p_t} \left[ \sum_{x \neq y} w_{x,y} c_r \frac{p_t(y)}{p_t(x)} \left| \frac{\bar{p}_t(x)}{\bar{p}_t(y)} - \frac{p_t(x)}{p_t(y)} \right|^2 \right]
$$
$$
= \mathcal{O}\left( \frac{\beta C_t^{\mathrm{unif}}(d + \log n)}{n p_0^{\min}} + \mathbb{P}(\mathcal{E}^c) C_t^{\mathrm{unif}} \log(C_t^{\mathrm{unif}}/\epsilon) \right).
$$

Finally, we integrate $\mathcal{L}_t(\widehat{s}_t)$ over $t \in [t_0, T]$ with $t_0 = 1/(\beta n)$ and $T = \log n/\beta$. Following the same computation in Appendix C.3, we can derive $\int_{t_0}^{T} C_t^{\mathrm{unif}} \mathrm{d}t \lesssim \beta + K$. On the other hand, applying Lemmas 3.4 and D.1 with the condition of $r_0$, we have $\int_{t_0}^{T} C_t^{\mathrm{unif}} \mathrm{d}t \lesssim r_0$. Therefore, we can conclude

$$
\int_{t_0}^{T} \mathcal{L}_t(\widehat{s}_t) \mathrm{d}t = \widetilde{\mathcal{O}}\left( \min\{r_0, \beta + K\} \frac{d}{n p_0^{\min}} + (\beta + K)\mathbb{P}(\mathcal{E}^c) \right).
$$

The proof is complete.

**Lemma D.1.** *Consider the conditional independent uniform process. For any time $t > 0$, any $i \in [d]$ and $y \in [K]^d$, we have*

$$
\mathbb{P}(X_0^i = y^i | X_t^{-i} = y^{-i}) = \sum_{A \subseteq [d] \setminus \{i\}} w_t^A(y) \mathbb{P}\left( X_0^i = y^i | E_A^{[d]\setminus\{i\}}(y) \right).
$$

*Here we denote the event $E_A^S(y) := \{X_0^A = y^A, X_0^j \neq y^j \text{ for any } j \in S \setminus A\}$ that $X_0$ agrees with $y$ on the index set $A \subseteq S$ and disagrees on every other index in $S$, and denote the weight $w_t^A(y)$ as*

$$
w_t^A(y) = \frac{\left( \frac{K\alpha_t}{1-\alpha_t} + 1 \right)^{|A|} \mathbb{P}\left( E_A^{[d]\setminus\{i\}}(y) \right)}{\sum_{B \subseteq [d]\setminus\{i\}} \left( \frac{K\alpha_t}{1-\alpha_t} + 1 \right)^{|B|} \mathbb{P}\left( E_B^{[d]\setminus\{i\}}(y) \right)}.
$$

*Proof.* We start with analyzing the probability $\mathbb{P}(X_0^i = y^i, X_t^{-i} = y^{-i})$. Utilizing the conditional independence and (6),

we have

$$\mathbb{P}(X_0^i = y^i, X_t^{-i} = y^{-i}) = \sum_{x_0:x_0^i=y^i} \prod_{j\neq i} \left( \alpha_t \mathbb{1}(x_0^j = y^j) + \frac{1-\alpha_t}{K} \right) p_0(x_0)$$

$$= \sum_{x_0:x_0^i=y^i} \left( \alpha_t + \frac{1-\alpha_t}{K} \right)^{I(x_0,y)} \left( \frac{1-\alpha_t}{K} \right)^{d-1-I(x_0,y)} p_0(x_0),$$

where we denote $I(x_0, y) = \left| \{j \neq i : x_0^j = y^j\} \right|$ as the number of identical entries in $x_0$ and $y$. Similarly, we have

$$\mathbb{P}(X_t^{-i} = y^{-i}) = \sum_{x_0} \left( \alpha_t + \frac{1-\alpha_t}{K} \right)^{I(x_0,y)} \left( \frac{1-\alpha_t}{K} \right)^{d-1-I(x_0,y)} p_0(x_0).$$

Therefore, we can derive the conditional probability $\mathbb{P}(X_0^i = y^i | X_t^{-i} = y^{-i})$ as

$$\mathbb{P}(X_0^i = y^i | X_t^{-i} = y^{-i}) = \frac{\mathbb{P}(X_0^i = y^i, X_t^{-i} = y^{-i})}{\mathbb{P}(X_t^{-i} = y^{-i})} = \frac{\sum_{x_0:x_0^i=y^i} \left( \frac{K\alpha_t}{1-\alpha_t} + 1 \right)^{I(x_0,y)} p_0(x_0)}{\sum_{x_0} \left( \frac{K\alpha_t}{1-\alpha_t} + 1 \right)^{I(x_0,y)} p_0(x_0)}.$$

Moreover, note that the collection $\left\{ E_A^{[d]\setminus\{i\}}(y) \right\}_{A\subseteq[d]\setminus\{i\}}$ forms a partition of the sample space, and we have $I(X_0, y) = |A|$ under $E_A^{[d]\setminus\{i\}}(y)$. Rewriting the summation over index sets yields

$$\mathbb{P}(X_0^i = y^i | X_t^{-i} = y^{-i}) = \frac{\mathbb{P}(X_0^i = y^i, X_t^{-i} = y^{-i})}{\mathbb{P}(X_t^{-i} = y^{-i})} = \frac{\sum_{A\subseteq[d]\setminus\{i\}} \left( \frac{K\alpha_t}{1-\alpha_t} + 1 \right)^{|A|} \mathbb{P}\left( X_0^i = y^i, E_A^{[d]\setminus\{i\}}(y) \right)}{\sum_{B\subseteq[d]\setminus\{i\}} \left( \frac{K\alpha_t}{1-\alpha_t} + 1 \right)^{|B|} \mathbb{P}\left( E_B^{[d]\setminus\{i\}}(y) \right)}.$$

Finally, we conclude the proof by applying $\mathbb{P}\left( X_0^i = y^i, E_A^{[d]\setminus\{i\}}(y) \right) = \mathbb{P}\left( X_0^i = y^i | E_A^{[d]\setminus\{i\}}(y) \right) \mathbb{P}\left( E_A^{[d]\setminus\{i\}}(y) \right)$. $\square$

**Lemma D.2.** *For any $\delta \in (0, 1)$ and given $y \in [K]^d$, it holds with probability at least $1 - \delta$ that*

$$\left| \widehat{\mathbb{P}}(X_0^S = y^S) - \mathbb{P}(X_0^S = y^S) \right| \leq \sqrt{\frac{3(\log |\mathcal{S}| + \log(2/\delta))}{n} \mathbb{P}(X_0^S = y^S)}, \quad \text{for any index set } S \subseteq \mathcal{S}.$$

*Proof.* By Lemma F.1, for any index set $S \subseteq \mathcal{S}$, we have

$$\mathbb{P}\left( \left| \frac{\widehat{\mathbb{P}}(X_0^S = y^S)}{\mathbb{P}(X_0^S = y^S)} - 1 \right| \geq \delta_S \right) \leq 2\exp\left( -\frac{1}{3}n\delta_S^2 \mathbb{P}(X_0^S = y^S) \right).$$

Setting $\delta_S = \sqrt{3\log(2|\mathcal{S}|/(\delta))/(n\mathbb{P}(X_0^S = y^S))}$ so that $2\exp\left( -n\delta_S^2 \mathbb{P}(X_0^S = y^S)/3 \right) = \delta/|\mathcal{S}|$ yields

$$\mathbb{P}\left( \left| \frac{\widehat{\mathbb{P}}(X_0^S = y^S)}{\mathbb{P}(X_0^S = y^S)} - 1 \right| \geq \delta_S, \text{ for any } S \in \mathcal{S} \right) \leq \delta.$$

In other words, it holds with probability that

$$\left| \widehat{\mathbb{P}}(X_0^S = y^S) - \mathbb{P}(X_0^S = y^S) \right| \leq \sqrt{\frac{3(\log(|\mathcal{S}| + \log(2/\delta))}{n} \mathbb{P}(X_0^S = y^S)},$$

for any index set $S \in \mathcal{S}$. $\square$

**Lemma D.3.** *Let $\mathcal{E} := \{y \in \mathcal{X} : P(X^S = y^S) \geq 12(d\log 2 + \log(2\delta))/n \text{ for any } S \subseteq [d] \text{ satisfying } P(X^S = y^S) > 0\}$. Denote $p_0^{\min} := \min\{P(X^S = y^S) : P(X^S = y^S) \neq 0, S \subseteq [d], y \in \mathcal{E}\}$. For any fixed $\delta \in (0,1)$, $i \in [d]$ and $(y^i, y^{-i}) \in [K]^d$, it holds with probability at least $1 - \delta$ that*

$$\left|\widehat{\mathbb{P}}(X_0^i = y^i | X_t^{-i} = y^{-i}) - \mathbb{P}(X_0^i = y^i | X_t^{-i} = y^{-i})\right| \leq \sqrt{\frac{c(d + \log(1/\delta))}{np_0^{\min}}}\mathbb{P}(X_0^i = y^i | X_t^{-i} = y^{-i}).$$

*Here $c > 0$ is some absolute constant.*

*Proof.* As shown in Lemma D.1, we can write the conditional probability $\mathbb{P}(X_0^i = y^i | X_t^{-i} = y^{-i})$ as

$$\mathbb{P}(X_0^i = y^i | X_t^{-i} = y^{-i}) = \sum_{A \subseteq [d]\backslash\{i\}} w_t^A(y) \mathbb{P}\left(X_0^i = y^i | E_A^{[d]\backslash\{i\}}(y)\right) = \frac{\sum_{A \subseteq [d]\backslash\{i\}} c_t^{|A|} p_{A,i}}{\sum_{A \subseteq [d]\backslash\{i\}} c_t^{|A|} p_A},$$

where we denote $c_t = K\alpha_t/(1 - \alpha_t) + 1$, $p_{A,i} = \mathbb{P}\left(X_0^i = y^i, E_A^{[d]\backslash\{i\}}(y)\right)$ and $p_A = \mathbb{P}\left(E_A^{[d]\backslash\{i\}}(y)\right)$, with event $E_A^S(y) := \{X_0^A = y^A, X_0^j \neq y^j \text{ for any } j \in S\backslash A\}$. For notational simplicity, we denote $\widehat{p}_{A,i} = \widehat{\mathbb{P}}\left(X_0^i = y^i, E_A^{[d]\backslash\{i\}}(y)\right)$ and $\widehat{p}_A = \widehat{\mathbb{P}}\left(E_A^{[d]\backslash\{i\}}(y)\right)$. We also denote

$$N_p = \sum_{A \subseteq [d]\backslash\{i\}} c_t^{|A|} p_{A,i}, \quad \text{and} \quad D_p = \sum_{A \subseteq [d]\backslash\{i\}} c_t^{|A|} p_A.$$

Note that $\mathbb{P}(X_0^i = y^i | X_t^{-i} = y^{-i})$ can be viewed as a function of $\{p_{A,i}\}_{A \subseteq [d]\backslash\{i\}}$ and $\{p_A\}_{A \subseteq [d]\backslash\{i\}}$. The first-order derivatives can be calculated as

$$\frac{\partial \mathbb{P}(X_0^i = y^i | X_t^{-i} = y^{-i})}{\partial p_{A,i}} = \frac{c_t^{|A|}}{D_p}, \quad \text{and} \quad \frac{\partial \mathbb{P}(X_0^i = y^i | X_t^{-i} = y^{-i})}{\partial p_A} = -\frac{c_t^{|A|} N_p}{D_p^2}.$$

When $p_0^{\min} \geq 12(d\log 2 + \log(2\delta))/n$, Lemma D.2 implies $p_A/2 \leq \widehat{p}_A \leq 2p_A$ and $p_{A,i}/2 \leq \widehat{p}_{A,i} \leq 2p_{A,i}$.

$$\left|\widehat{\mathbb{P}}(X_0^i = y^i | X_t^{-i} = y^{-i}) - \mathbb{P}(X_0^i = y^i | X_t^{-i} = y^{-i})\right| \leq \sum_{A \subseteq [d]\backslash\{i\}} \frac{2c_t^{|A|}}{D_p}|\widehat{p}_{A,i} - p_{A,i}| + \sum_{A \subseteq [d]\backslash\{i\}} \frac{8c_t^{|A|} N_p}{D_p^2}|\widehat{p}_A - p_A|.$$

Take $\mathcal{S} = \{A \subseteq [d] \backslash \{i\} : p_A > 0\}$ in Lemma D.2. The cardinality $\mathcal{S}$ is naturally bounded by $|\mathcal{S}| \leq 2^d$. This yields that with probability at least $1 - \delta$,

$$|\widehat{p}_{A,i} - p_{A,i}| \leq \sqrt{\frac{3(d\log 2 + \log(2/\delta))}{n}p_{A,i}}, \quad \text{and} \quad |\widehat{p}_A - p_A| \leq \sqrt{\frac{3(d\log 2 + \log(2/\delta))}{n}p_A}.$$

Moreover, we can derive

$$\left|\widehat{\mathbb{P}}(X_0^i = y^i | X_t^{-i} = y^{-i}) - \mathbb{P}(X_0^i = y^i | X_t^{-i} = y^{-i})\right| \leq \sqrt{\frac{3(d\log 2 + \log(2/\delta))}{n}} \sum_{A \subseteq [d]\backslash\{i\}} \frac{2c_t^{|A|}\sqrt{p_{A,i}}}{D_p} + \frac{8c_t^{|A|}\sqrt{p_A}N_p}{D_p^2}.$$

Since $N_p/D_p = \mathbb{P}(X_0^i = y^i | X_t^{-i} = y^{-i})$, we further have

$$\frac{\left|\widehat{\mathbb{P}}(X_0^i = y^i | X_t^{-i} = y^{-i}) - \mathbb{P}(X_0^i = y^i | X_t^{-i} = y^{-i})\right|}{\mathbb{P}(X_0^i = y^i | X_t^{-i} = y^{-i})} \leq \sqrt{\frac{12(d\log 2 + \log(2/\delta))}{n}} \sum_{A \subseteq [d]\backslash\{i\}} \frac{c_t^{|A|}\sqrt{p_{A,i}}}{N_p} + \frac{4c_t^{|A|}\sqrt{p_A}}{D_p}.$$

According to the condition $P(X^S = y^S) > p_0^{\min}$ for $S \subseteq [d]$ satisfying $P(X^S = y^S) > 0$, we obtain

$$\left| \widehat{\mathbb{P}}(X_0^i = y^i | X_t^{-i} = y^{-i}) - \mathbb{P}(X_0^i = y^i | X_t^{-i} = y^{-i}) \right| \leq 10 \sqrt{\frac{3(d \log 2 + \log(2/\delta))}{n p_0^{\min}}} \mathbb{P}(X_0^i = y^i | X_t^{-i} = y^{-i}).$$

The proof is complete. $\qquad\qquad\qquad\qquad\qquad\qquad\qquad\qquad\qquad\qquad\qquad\qquad\qquad\qquad\qquad\square$

### D.2. Proof of Lemma 4.2

Consider the distribution $\bar{P}_t$, which is generated by the forward process (1) with the absorbing transition matrix (Lemma 3.5) and initialized at the empirical distribution $\bar{P}_0$ with probability mass $\bar{p}_0(x) = \frac{1}{n} \sum_{k=1}^n \mathbb{1}\{x = x^{(k)}\}$ of the dataset $\{x^{(k)}\}_{k=1}^n$. Lemma 3.5 implies that the discrete scores estimated from the empirical data distribution can be formulated as

$$\frac{\bar{p}_t(x)}{\bar{p}_t(y)} = \frac{\alpha_t}{1 - \alpha_t} \widehat{\mathbb{P}}(X_0^i = x^i | X_0^A = y^A), \quad \text{where } A = \{j \in [d] : y^j \neq \mathsf{M}\},$$

where $\widehat{\mathbb{P}}$ denotes the probability measure induced by sampling $X_0$ from the empirical distribution $\bar{P}_0$.

The key to upper bound $\mathbb{E}[\mathcal{L}_t(\widehat{s}_t)]$ is the concentration property of $\widehat{\mathbb{P}}(X_0^i = x^i | X_0^A = y^A)$. Taking $\mathcal{S} = \{A, A \cup \{i\}\}$ in Lemma D.2 yields that at probability at least $1 - 1/n$

$$\left| \frac{\bar{p}_t(x)}{\bar{p}_t(y)} - \frac{p_t(x)}{p_t(y)} \right| = \frac{\alpha_t}{1 - \alpha_t} \left| \widehat{\mathbb{P}}(X_0^i = x^i | X_0^A = y^A) - \mathbb{P}(X_0^i = x^i | X_0^A = y^A) \right|$$

$$= \mathcal{O}\left( \frac{\alpha_t}{1 - \alpha_t} \sqrt{\frac{\log n}{n}} \mathbb{P}(X_0^i = x^i | X_0^A = y^A) \right).$$

Moreover, when $\mathbb{P}(X_0^i = x^i | X_0^A = y^A) \geq c \log n / n$ for some constant $c > 0$, the above result also implies

$$c_r^{-1} \frac{p_t(x)}{p_t(y)} \leq \frac{\bar{p}_t(x)}{\bar{p}_t(y)} \leq c_r \frac{p_t(x)}{p_t(y)},$$

where $c_r \geq 1$ is a constant. Furthermore, we can bound the score estimation error by

$$\mathcal{L}_t(\widehat{s}_t) \leq \mathbb{E}\left[ \mathbb{E}_{y \sim p_t} \left[ \sum_{x \neq y} w_{x,y} c_r \frac{p_t(y)}{p_t(x)} \left| \frac{\bar{p}_t(x)}{\bar{p}_t(y)} - \frac{p_t(x)}{p_t(y)} \right|^2 \mathbb{1}\left( \frac{p_t(x)}{p_t(y)} \geq \frac{c\alpha_t \log n}{(1 - \alpha_t)n} \right) \right] \right]$$

$$+ \mathbb{E}\left[ \mathbb{E}_{y \sim p_t} \left[ \sum_{x \neq y} w_{x,y} \epsilon^{-1} \left| \frac{\bar{p}_t(x)}{\bar{p}_t(y)} - \frac{p_t(x)}{p_t(y)} \right|^2 \mathbb{1}\left( \frac{p_t(x)}{p_t(y)} < \frac{c\alpha_t \log n}{(1 - \alpha_t)n} \right) \right] \right]$$

$$= \mathcal{O}\left( \frac{\beta \alpha_t \log n}{(1 - \alpha_t)n} \left( 1 + \frac{\log n}{n\epsilon} \right) \mathcal{K}_t \right),$$

where $\mathcal{K}_t := \sum_{y : p_y(y) > 0} p_t(y) \left| \{x : x^i \neq \mathsf{M}, y^i = \mathsf{M}, x^{-i} = y^{-i} \text{ for some } i, \text{ and } p_t(x) > 0\} \right|$. Since we take $\epsilon = c_0 \log n / n$, we can further derive

$$\int_{t_0}^T \mathcal{L}_t(\widehat{s}_t^{\text{absorb}}) dt = \widetilde{\mathcal{O}}\left( \frac{\mathcal{K}^{\max}}{n} \right),$$

where $\mathcal{K}^{\max} := \max_{t \in [t_0, T]} \mathcal{K}_t$.

# E. Experimental Details

**Data Generation.** We generate synthetic sequential data from Markov chains over vocabulary size $K = 100$ and sequence length $d = 12$. Each Markov chain is defined by a $K \times K$ transition matrix $Q^*$ with controlled sparsity structure. For each state $i \in [K]$, we designate $S$ neighboring states as the significant support. The significant support receives total probability mass $1 - \epsilon_{\text{bg}}$ with $\epsilon_{\text{bg}} = 0.01$, where each significant state is guaranteed minimum probability $p_0^{\min}$. The remaining mass within the significant support is distributed according to a mixture of uniform and exponential distributions. Background states share the residual mass $\epsilon_{\text{bg}}$ uniformly. We sample sequences from the stationary distribution of the resulting Markov chain.

**Training Details.** We train transformer-based discrete score networks following the SEDD architecture (Lou et al., 2023): 4-layer transformer with hidden dimension 128 and 4 attention heads. Both absorbing and uniform forward processes use identical hyperparameters: learning rate $5 \times 10^{-4}$ with AdamW optimizer ($\beta_1 = 0.9$, $\beta_2 = 0.999$, weight decay 0.001), batch size 128, and 20,000 training iterations. Training data consists of $2.5 \times 10^5$ sequences sampled from the Markov chain.

**Evaluation.** For each trained model, we generate 10,000 samples using 128 Euler discretization steps. From these samples, we estimate the empirical transition matrix $\widehat{Q}$ by counting consecutive token pairs:

$$\widehat{Q}_{ij} = \frac{\#\{(x_t, x_{t+1}) = (i, j)\}}{\#\{x_t = i\}}.$$

We report the normalized $\ell_1$ distance $\mathcal{D}(\widehat{Q}, Q^*) = \|\widehat{Q} - Q^*\|_1/(2K)$, which lies in $[0, 1]$ and equals zero if and only if $\widehat{Q} = Q^*$. All experiments are repeated over 10 random seeds, and we report mean $\pm$ standard deviation.

# F. Auxiliary Lemmas

**Lemma F.1.** *Let $X_1, \ldots, X_n$ i.i.d. $\sim \mathrm{Cat}(p_0)$ and $\bar{p}_0 = \frac{1}{n} \sum_{k=1}^n X_k$. Denote $[\bar{p}_t] = \exp(\int_0^t Q_\tau \, d\tau)\bar{p}_0$. Then for any $i \in [K]$ and $\delta \in (0, 1]$, we have*

$$\mathbb{P}\left(\frac{[\bar{p}_t]_i}{[p_t]_i} - 1 \geq \delta\right) \leq \exp\left(-\frac{3}{8} n\delta^2 [p_t]_i\right) \quad and \quad \mathbb{P}\left(\frac{[\bar{p}_t]_i}{[p_t]_i} - 1 \leq -\delta\right) \leq \exp\left(-\frac{3}{8} n\delta^2 [p_t]_i\right).$$

*Proof.* We focus on bounding $\mathbb{P}\left(\frac{[\bar{p}_t]_i}{[p_t]_i} - 1 \geq \delta\right)$. The bound for $\mathbb{P}\left(\frac{[\bar{p}_t]_i}{[p_t]_i} - 1 \leq -\delta\right)$ follows a similar argument. For $\lambda \in (0, 3n)$, we first rewrite the probability as

$$\mathbb{P}\left(\frac{[\bar{p}_t]_i}{[p_t]_i} - 1 \geq \delta\right) = \mathbb{P}\left([\bar{p}_t]_i - [p_t]_i \geq \delta[p_t]_i\right)$$

$$= \mathbb{P}\left(\exp\left(\lambda([\bar{p}_t]_i - [p_t]_i)\right) \geq \exp\left(\lambda\delta[p_t]_i\right)\right)$$

$$\leq \frac{\mathbb{E}\exp\left(\lambda([\bar{p}_t]_i - [p_t]_i)\right)}{\exp\left(\lambda\delta[p_t]_i\right)},$$

where the last inequality is due to Markov inequality. Now we denote

$$g_k = e_i^\top \bar{Q}_t (X_k - p_0),$$

where $\bar{Q}_t := \exp(\int_0^t Q_\tau \, d\tau)$. Since $X_k \sim \mathrm{Cat}(p_0)$, we have $\mathbb{E}X_k = p_0$ and $\mathrm{Var}(X_k) = \mathrm{diag}(p_0) - p_0 p_0^\top$. This further gives

$$\mathbb{E}g_k = \mathbb{E}e_i^\top \bar{Q}_t (X_k - p_0) = 0. \tag{24}$$

Recall $[e_i^\top \bar{Q}_t]_j = P_{j,i}(0, t)$ and $[p_t]_i = e_i^\top \bar{Q}_t p_0$. We can derive

$$
\begin{aligned}
\operatorname{Var}(g_k) &= e_i^\top \bar{Q}_t \operatorname{Var}(X_k) \bar{Q}_t^\top e_i \\
&= e_i^\top \bar{Q}_t \operatorname{diag}(p_0) \bar{Q}_t^\top e_i - (e_i^\top \bar{Q}_t p_0)^2 \\
&\leq [p_t]_i - [p_t]_i^2.
\end{aligned}
\tag{25}
$$

Next, we bound $\mathbb{E} \exp\left(\lambda([\bar{p}_t]_i - [p_t]_i)\right)$ using the first and second moments of $g_k$. We can rewrite the expectation as

$$
\mathbb{E} \exp\left(\lambda([\bar{p}_t]_i - [p_t]_i)\right) = \mathbb{E} \exp\left(\frac{\lambda}{n} \sum_{k=1}^n e_i^\top \bar{Q}_t(X_k - p_0)\right) \overset{(i)}{=} \prod_{k=1}^n \mathbb{E} \exp\left(\frac{\lambda}{n} g_k\right).
\tag{26}
$$

Here (i) applies the independence of $X_k$'s. Moreover, noting that $g_k \in [-1, 1]$, we have

$$
\begin{aligned}
\mathbb{E} \exp\left(\frac{\lambda}{n} g_k\right) &\leq \mathbb{E}\left[1 + \frac{\lambda}{n} g_k + \frac{\lambda^2}{n^2} g_k^2 \sum_{l=2}^\infty \frac{\lambda^{l-2}}{l! n^{l-2}}\right] \\
&\leq \mathbb{E}\left[1 + \frac{\lambda}{n} g_k + \frac{\lambda^2}{2n^2} g_k^2 \sum_{l=2}^\infty \left(\frac{\lambda}{3n}\right)^{l-2}\right] \\
&= 1 + \frac{\lambda}{n} \mathbb{E} g_k + \frac{\lambda^2}{2n^2} \cdot \frac{\mathbb{E} g_k^2}{1 - \frac{\lambda}{3n}} \\
&= 1 + \frac{\lambda^2 [p_t]_i(1 - [p_t]_i)}{2n(n - \lambda/3)}.
\end{aligned}
$$

Applying $1 + z \leq \exp(z)$, we can further get

$$
\mathbb{E} \exp\left(\frac{\lambda}{n} g_k\right) \leq \exp\left(\frac{[p_t]_i(1 - [p_t]_i)}{2n(n - \lambda/3)}\right).
$$

Plug the above inequality into (26), then we obtain

$$
\mathbb{E} \exp\left(\lambda([\bar{p}_t]_i - [p_t]_i)\right) \leq \prod_{k=1}^n \exp\left(\frac{\lambda^2 [p_t]_i(1 - [p_t]_i)}{2n(n - \lambda/3)}\right).
$$

This gives

$$
\mathbb{P}\left(\frac{[\bar{p}_t]_i}{[p_t]_i} - 1 \geq \delta\right) \leq \frac{\mathbb{E} \exp\left(\lambda([\bar{p}_t]_i - [p_t]_i)\right)}{\exp\left(\lambda\delta[p_t]_i\right)} \leq \exp\left(\frac{\lambda^2 [p_t]_i(1 - [p_t]_i)}{2(n - \lambda/3)} - \lambda\delta[p_t]_i\right).
$$

Now we take $\lambda = 3n\delta/4$, which yields

$$
\mathbb{P}\left(\frac{[\bar{p}_t]_i}{[p_t]_i} - 1 \geq \delta\right) \leq \exp\left(\frac{\frac{9}{16}n^2\delta^2[p_t]_i(1 - [p_t]_i)}{2(n - n\delta/4)} - \frac{3}{4}n\delta^2[p_t]_i\right) \leq \exp\left(-\frac{3}{8}n\delta^2[p_t]_i\right).
$$

Likewise, we can write the other probability

$$
\mathbb{P}\left(\frac{[\bar{p}_t]_i}{[p_t]_i} - 1 \leq -\delta\right) = \mathbb{P}\left([p_t]_i - [\bar{p}_t]_i \geq \delta[p_t]_i\right).
$$

Due to symmetry, we can follow the same argument and conclude $\mathbb{P}\left(\frac{[\bar{p}_t]_i}{[p_t]_i} - 1 \leq -\delta\right) \leq \exp\left(-\frac{3}{8}n\delta^2[p_t]_i\right)$. $\qquad\square$

**Lemma F.2.** *Let $X_1, \ldots, X_n$ i.i.d. $\sim \operatorname{Cat}(p_0)$ and $\bar{p}_0 = \frac{1}{n}\sum_{k=1}^n X_k$. Denote $[\bar{p}_t] = \exp(\int_0^t Q_\tau \, d\tau)\bar{p}_0$. Then for any*

$i \in [K]$ *and* $\delta \in (0, 1]$*, it holds with probability at least* $1 - \delta$ *that*

$$|[\bar{p}_t]_i - [p_t]_i| \leq \sqrt{\frac{4[p_t]_i(1 - [p_t]_i)\log\frac{2}{\delta}}{n}} + \frac{4\log\frac{2}{\delta}}{3n}.$$

*Proof.* Lemma F.2 is a direct application of Lemma F.3. Denote $g_k = e_i^\top \bar{Q}_t(X_k - p_0)$, where $\bar{Q}_t := \exp(\int_0^t Q_\tau \, d\tau)$. Then we have $[\bar{p}_t]_i - [p_t]_i = \frac{1}{n}\sum_{k=1}^n g_k$. Note that $g_1, \ldots, g_n$ are i.i.d. random variables in $[-1, 1]$, and by (24) and (25), we have

$$\mathbb{E}g_k = 0, \quad \text{and} \quad \text{Var}(g_k) \leq [p_t]_i(1 - [p_t]_i).$$

Applying Lemma F.3 on $g_1, \ldots, g_n$, we can get

$$|[\bar{p}_t]_i - [p_t]_i| = \left|\frac{1}{n}\sum_{k=1}^n g_k\right| \leq \sqrt{\frac{4[p_t]_i(1 - [p_t]_i)\log\frac{2}{\delta}}{n}} + \frac{4\log\frac{2}{\delta}}{3n},$$

which holds with probability at least $1 - \delta$. □

Lemma F.3 is a direct result of Bernstein's inequality.

**Lemma F.3.** *Let* $X_1, \ldots, X_n$ *be i.i.d. random variables, and* $\forall i$*, we have* $|X_i - \mathbb{E}X_i| \leq R$*. Let* $\mu = \mathbb{E}[X_i]$ *and* $\sigma^2 = \text{Var}[X_i]$*. Then with probability* $1 - \delta$*, we have*

$$\left|\frac{1}{n}\sum_{i=1}^n X_i - \mu\right| \leq \sqrt{\frac{4\sigma^2\log\frac{2}{\delta}}{n}} + \frac{4R\log\frac{2}{\delta}}{3n}.$$

