# OpenReview forum: "Generalization Bounds for Discrete Diffusion: Statistical Advantage of Masking"
_ICML.cc/2026/Conference — ICML 2026 regular_

### Official Review · Reviewer_cUPt · 2026-03-03

**Soundness:** 3
**Presentation:** 3
**Significance:** 4
**Originality:** 4
**Overall Recommendation:** 6
**Confidence:** 4

**Summary:**

In this work, the forward process of a discrete diffusion model is investigated in the regime where the diffusion model is trained to convergence (learning the empirical data distribution) on finite sample complexity. Two schemes are compared: one in which the data distribution is gradually corrupted by uniform replacement, and one where the corruption is achieved by randomly replacing parts of the data vector with a masking token. The authors investigate how the TV distance between the distribution sampled by the model and the data distribution behaves under the two corruption procedures. They find that this distance scales with the space occupied by the corrupted distributions, which spans the whole ambient space in the case of uniform replacement, and for the masking procedure is upper bounded by the domain of the data distribution. Thus they show that masking has a statistical advantage, because the domain over which the score must be learned is smaller.

**Compliance With Llm Reviewing Policy:**

Affirmed.

**Final Justification:**

The authors have answered all my questions. I maintain my score.

**Key Questions For Authors:**

C_t is an upper bound to the score, and scales the contribution to the TV distance which occurs due to finite sample complexity. In the uniform case, for large t, this upper bound should trend to one, simply because the large t distribution is quasi-uniform. This is also visible from Lemma 3.4. For the masking process, it is a bit intuitive how the bound on C_t in Lemma 3.5 arises. Intuitively, in the masking procedure, probability mass is shuffled from the empirical distribution onto data vectors with increasing number of masked tokens. That means that the fraction \bar{p}_t(x)/\bar{p}_t(y) can become very large when t is large, x has many masked tokens and y only very few masked tokens. The bound in Lemma 3.5 appears to show the opposite behavior, namely decreasing with t. Can the authors comment on how this bound comes about, intuitively and why it does not suffer from this effect?

**Limitations:**

yes

**Strengths And Weaknesses:**

Strengths

I found this manuscript a pleasure to read; the results are clearly presented and validated, the conclusions drawn are intuitive, and the contrast between the uniform and masking corruption procedure is tangible. The authors clearly show that
- masking is a superior corruption technique based on the reduced domain of the function one needs to learn
- the bounds on the TV distance scale as expected with the domain of this function
- this result explains the superiority of masking forward processes observed empirically

Weaknesses

I have only one minor confusion regarding the constant C_t, explained below.

---

> ### Author Rebuttal · Authors · 2026-03-31
>
> Thank you for the enthusiastic assessment and for the kind words on the clarity and contribution of our work! We are glad that the comparison between uniform and masking corruption resonated. We would like to answer your question below.
>
> ***Q: Why does $C_t^{\text{absorb}}$ decrease with $t$, given that probability mass concentrates on heavily masked sequences at large $t$?***
>
> This is a great question. The key point is that $C_t$ in our analysis bounds only local score ratios along admissible reverse transitions, i.e., pairs $(x, y)$ satisfying $p_t(y)Q_t(y,x) > 0$. For the absorbing process with transition matrix in Equation (9), such a transition exists only when $x$ and $y$ differ at exactly one coordinate $i$, with $y^i = M$, $x^i \neq M$, and $x^{-i} = y^{-i}$. In other words, $x$ has exactly one fewer mask than $y$. For these local pairs, Lemma 3.5 gives $p_t(x)/p_t(y) = \frac{\alpha_t}{1-\alpha_t} P(X_0^i = x^i | X_0^A = y^A),$ which indeed decreases with $t$ because $\alpha_t/(1-\alpha_t)$ shrinks as masking increases.
>
> Your intuition about large ratios between states with very different numbers of masks is correct at a global level: arbitrary state pairs can certainly have large probability ratios. However, those global ratios do not enter the backward generator or the score-entropy objective. What matters for the reverse dynamics is only the local edge-wise ratio between states connected by a single unmasking step, and these local ratios are naturally controlled by the $\alpha_t/(1-\alpha_t)$ factor. This is a structural advantage of the absorbing process: its unidirectional transitions ensure that the relevant score ratios remain well-behaved, in contrast to the uniform process where bidirectional transitions keep ratios bounded away from zero even at large $t$.
>
> We appreciate your support and are happy to discuss further.

---

> > ### Author Rebuttal · Reviewer_cUPt · 2026-04-01
> >
> > Thank you for answering my question. I have no further concerns.

---

> > > ### Author Response · Authors · 2026-04-02
> > >
> > > Thank you for the acknowledgment! We greatly appreciate your feedback.

---

### Official Review · Reviewer_mt1B · 2026-03-04

**Soundness:** 2
**Presentation:** 2
**Significance:** 3
**Originality:** 3
**Overall Recommendation:** 4
**Confidence:** 3

**Summary:**

This paper analyzes the statistical properties of general and specific discrete diffusion models. Concretely, it builds a framework of TV bounds based on the score entropy analysis, which reveals important interpretable factors affecting the bounds. Based on it, this paper analyzes different prevalent diffusion models, and explains the superiority of masking over uniform diffusion.

**Compliance With Llm Reviewing Policy:**

Affirmed.

**Key Questions For Authors:**

Questions

- How is the proposition 2 in work [1] leveraged to prove the proposition 3.1 in this work?
- What is the difference between $\tilde{p}$ and $\hat{p}$ in the proof of the proposition 3.1?
- In the lemma 3.2, what are the omitted logarithmic factors (in line 796), how will the omitted factors affect the result? Also, how is the result in line 796 simplified from its above derivations?
- In addition to the tightness on $|\mathcal{X}|$, how is the overall tightness of the bound in the lemma 3.2?
- How do the assumption of $\beta_t \equiv \beta$ in lemma 3.6 and 3.7 affects the results of this paper?


[1] Chen, H. and Ying, L. Convergence analysis of discrete diffusion model: Exact implementation through uniformization. arXiv preprint arXiv:2402.08095, 2024.

**Limitations:**

yes

**Strengths And Weaknesses:**

Strengths

- This work provides a statistical framework, pointing out that the value range of the discrete score and the support size of noisy data are the two important factors affecting the TV bound of discrete diffusion models.
- Based on the two factors, this work analyzes the theoretical advantages of absorbing over uniform diffusion processes, and provides theoretical insights into the design of the discrete diffusion forward kernel.
- By considering the properties of natural language, the bounds are further refined for uniform and absorbing state diffusion models.

Weaknesses

- Some derivations omit many intermediate steps, making them hard to follow. Please see the questions section for details.
- The overall tightness of the TV bound in the lemma 3.2 seems unclear. My concern is that a lower but loose upper bound cannot guarantee the error is also lower.
- The assumption of constant noise schedule $\beta_t \equiv \beta$ used in lemma 3.6 and 3.7 seems too strong. In practice, $\beta_t$ is far from a constant, e.g. in a typical log-linear schedule, it is $\frac{1}{1-t} \in [1, +\infty)$.

---

> ### Author Rebuttal · Authors · 2026-03-31
>
> Thank you for the detailed reading and for recognizing the key insight of our framework. We would like to address each concern below.
>
> ***W1: Some derivations omit intermediate steps, hard to follow.***
>
> We will add more intermediate steps in the revision to improve readability. We would address the specific proof details below.
>
> ***W2: Overall tightness of the TV bound in Lemma 3.2 is unclear. A smaller upper bound does not by itself guarantee a smaller true error.***
>
> This is a great question that we address from two angles. First, as noted in Remark 3.3, the scaling $|\mathcal{X}|/n$ in Lemma 3.2 matches the minimax rate $\Theta(\sqrt{|\mathcal{X}|/n})$ for discrete distribution estimation in TV distance (Han et al., 2015). This means that over the class of all distributions on $\mathcal{X}$ **without further structural assumptions**, no estimator can improve upon the $|\mathcal{X}|/n$ scaling. The gap between absorbing and uniform thus reflects a genuine difference in effective estimation dimension — absorbing operates over $|\mathcal{X}^{\text{data}}|$ while uniform must cover $K^d$—rather than an artifact of loose analysis.
>
> Second, the comparative conclusion is robust because the separation arises from structural differences (support preservation vs. expansion, decaying vs. non-decaying $C_t$), not from looseness in the bounding technique. Our empirical results (Tables 1–2) precisely match the bound behavior: absorbing outperforms uniform when sparsity is high (small $S$), and the gap diminishes as $S$ increases. Tightening the $C_t$ dependence is valuable future work.
>
> ***W3: Constant noise schedule $\beta_t \equiv \beta$ is too strong for practice.***
>
> The constant schedule is standard in theoretical analyses of discrete diffusion [1]. Our results can extend to general schedules: the core estimation error bound in Lemma 3.2 holds for arbitrary $Q_t$ with **any** $\beta_t$. The constant schedule is only used in Lemmas 3.6–3.7 to obtain closed-form expressions for $I_{[t_0,T]}$. For a general $\beta_t$, the integrals $\int_{t_0}^{T} \frac{\beta_t \alpha_t}{1-\alpha_t} dt$ (absorbing) and $\int_{t_0}^{T} \beta_t(1 + \frac{K\alpha_t}{1-\alpha_t}) dt$ (uniform) would replace the closed-form expressions (Appendix C). The qualitative comparison is schedule-independent: for any $\beta_t$ s.t. $\alpha_t \to 0$, the absorbing integral benefits from the vanishing $\alpha_t/(1-\alpha_t)$, while the uniform integral always retains  the $\int_{t_0}^T \beta_t dt$ term that does not vanish. We will add this discussion in the revision.
>
> ***Q1: How is Proposition 2 in work [1] leveraged to prove Proposition 3.1 in this work?***
>
> We use their Proposition 2 to bound $\text{KL}(P^\leftarrow_{t} | \tilde{P}^\leftarrow_{t} ) \leq \int_{t_0}^{T} \mathcal{L}_t(s_t) dt$  (Appendix B.1), then apply Pinsker's inequality to obtain the TV bound in the estimation error term. This connects the score matching objective (Lemma 3.2) to distributional estimation error.
>
> ***Q2: What is the difference between $\tilde{p}$ and $\hat{p}$ in the proof of Proposition 3.1?***
>
> In the proof of Proposition 3.1, $\tilde{P}^\leftarrow_{t}$ denotes the backward process starting from the true terminal distribution $P_T$ with the approximate score $s_t$ (Line 643, Appendix B.1), while $\hat{P}^\leftarrow_t$ denotes the backward process starting from the stationary distribution $P_\infty$ with the same approximate score. The difference between them captures the mixing error from replacing $P_T$ with $P_\infty$. We will clarify this notation in the revision.
>
> ***Q3: In the lemma 3.2, what are the omitted logarithmic factors (in line 796), how will the omitted factors affect the result? Also, how is the result in line 796 simplified from its above derivations?***
>
> The $\tilde{O}(\cdot)$ in Lemma 3.2 hides factors of $\log(C_t/\epsilon)$ and $\log(2n)$ (Line 796, Appendix B.3). Since the dominant separation between kernels arises from the polynomial dependence on $C_t$ and on the support size, the logarithmic factors do not affect the qualitative scaling comparison. The simplification from the preceding derivation aggregates the three cases (equations 16, 17, 18) by upper-bounding each case's Bregman divergence contribution and summing over all state pairs $(x,y)$ with $p_t(y)>0$.
>
> ***Q4: In addition to the tightness on $|\mathcal{X}|$, how is the overall tightness of the bound in the lemma 3.2?***
>
> See ***W2*** above. The $1/n$ rate is also optimal, matching the minimax rate for discrete distributions estimation. The remaining factor is $C_t$; we conjecture it is near-tight based on the connection to the Bregman divergence analysis, but a formal lower bound is left as future work.
>
> ***Q5: How does $\beta_t \equiv \beta$ affect the results of this paper?***
>
> See our response to W3 above.
>
> We hope our responses address your concerns. We will incorporate expanded proof details, general noise schedule discussion, and improved notation in the next version.

---

> > ### Author Rebuttal · Reviewer_mt1B · 2026-04-01
> >
> > Thank you for your response. I think most concerns have been addressed; the only remaining one is the tightness on $C_t$, which is fine to leave for future work. I will raise my score to 4.

---

> > > ### Author Response · Authors · 2026-04-02
> > >
> > > Thank you for the follow-up and for raising your score. We appreciate your careful engagement with the technical details, and we are glad that our rebuttal addressed most of your concerns. We will make the discussion of $C_t$ more explicit in the revision.

---

### Official Review · Reviewer_4cnz · 2026-03-12

**Soundness:** 3
**Presentation:** 3
**Significance:** 2
**Originality:** 2
**Overall Recommendation:** 4
**Confidence:** 2

**Summary:**

This paper investigates the theoretical underpinnings of why masked (absorbing) discrete diffusion models empirically outperform uniform discrete diffusion models. It represents a step toward bridging the gap between empirical observations and theoretical understanding from a sample complexity perspective. Overall, the authors discuss a central concept regarding how the forward corruption kernel governs generalization. The core finding is that the sample complexity of different discrete diffusion models heavily depends on the volume of the support sets of the marginal distributions along the forward trajectory. Because the uniform forward transition expands the effective data support to the entire high-dimensional discrete space, it incurs a much larger sample complexity than the absorbing forward transition, which preserves support sparsity. Overall, this work outlines an important concept that explains the statistical advantages of masked diffusion models.

**Compliance With Llm Reviewing Policy:**

Affirmed.

**Ethical Review Flag:**

Flag this paper for an ethics review.

**Final Justification:**

The rebuttal has partially resolved my concerns

**Key Questions For Authors:**

Please check the weaknesses part.

**Limitations:**

This is a theoretical paper; no significant social consequences must be highlighted.

**Strengths And Weaknesses:**

**Strengths:**
1. Relevance and Motivation: Studying the superior performance of masked discrete diffusion from a sample complexity perspective is a highly relevant and interesting problem. The paper makes commendable progress in closing the gap between empirical practice and statistical learning theory.
2. Clear and Insightful Qualitative Conclusions: The theoretical results are presented clearly. Although the sample complexity upper bounds exhibit exponential dependence on the sequence length (dimension), the qualitative insights are valuable. Specifically, the implication that the advantage of the absorbing process diminishes as the density of the effective data support increases is both intuitive and well-supported.
3. Empirical Validation: The authors successfully corroborate their qualitative theoretical conclusions with well-designed synthetic numerical experiments.

**Weaknesses:**
1. Loose Sample Complexity Bounds: The derived upper bounds for sample complexity appear to be quite loose. Specifically, the authors provide an $\tilde{\mathcal{O}}(2^{d-1}|\mathcal{X}^{\text{data}}|/n)$ complexity for masked discrete diffusion. This implies that achieving a small expected score entropy requires a sample size that scales exponentially with the dimension $d$, which contradicts practical observations where these models scale efficiently. The authors might consider adopting more concrete structural assumptions or continuous embeddings, similar to the techniques used in [1], to achieve tighter, dimension-independent (or polynomial) bounds.
2. Limited Technical Novelty: The technical challenges overcome in the proofs seem somewhat limited. Building upon existing minimax limits, the core analysis can largely be viewed as a natural extension of the standard minimax rate for discrete distribution estimation applied to the discrete diffusion loss. The paper would benefit from highlighting any unique technical hurdles overcome specifically for the diffusion trajectory.
3. Missing Related Work: The literature review is missing a few highly relevant recent works regarding the sampling complexity of masked/absorbing discrete diffusion. Specifically, [2] provides convergence and complexity guarantees for Euler and uniformization samplers in this context. Furthermore, [3] explains the faster convergence of masked/absorbing discrete diffusion compared to uniform diffusion from the perspective of the total outgoing rate. Discussing these works would provide a more comprehensive view of the current theoretical landscape.


References:

[1] Wakasugi S, Suzuki T. State Size Independent Statistical Error Bound for Discrete Diffusion Models. The Thirty-ninth Annual Conference on Neural Information Processing Systems.

[2] Liang Y, Huang R, Lai L, et al. Absorb and Converge: Provable Convergence Guarantee for Absorbing Discrete Diffusion Models. The Thirty-ninth Annual Conference on Neural Information Processing Systems.

[3] Huang X, Lin Y, Jain N, et al. On the Complexity Theory of Masked Discrete Diffusion: From $\mathrm{poly}(1/\epsilon)$ to Nearly $\epsilon$-Free. arXiv preprint arXiv:2509.21835, 2025.

---

> ### Author Rebuttal · Authors · 2026-03-31
>
> Thank you for the careful reading and for accurately identifying our core contributions. We would like to reply to your concerns and suggestions below.
>
> ***W1: Sample complexity bounds are exponential in $d$, appearing loose. Suggests continuous embedding assumptions in Wakasugi & Suzuki (2025).***
>
> We agree that Theorem 3.8 carries worst-case exponential prefactors in the sequence length $d$. This is a deliberate consequence of our scope: we ask a comparative question: *does the choice of forward kernel change sample complexity scaling?* The separation is clear: absorbing scales with $|\mathcal{X}^{\text{data}}|$ vs. $K^d$ for uniform. This result is established without any distributional assumption such as embedding or smoothness, and the exponential prefactors are the cost of this generality.
>
> Importantly, our paper does **not** stop at Theorem 3.8. Section 4 shows that both kernels further adapt to finer distributional structure through distinct mechanisms. For the absorbing process, Lemma 4.2 yields $\tilde{O}(K_{\max}/n)$, where $K_{\max}$ can be dramatically smaller than the vocab size $K$ when only a small subset of tokens are valid completions at each masked position (Lines 397-398). For uniform, Lemma 4.1 yields $\tilde{O}(r_0 d/(n p^\min_0))$ when probability mass concentrates within the significant set $\mathcal{E}$. Both refinements lead to substantially tighter bounds than Theorem 3.8.
>
> Regarding Wakasugi & Suzuki (2025):  their assumptions (symmetric transition matrix, scores bounded away from zero) are not directly compatible with the absorbing kernel, which is non-reversible and naturally admits zero scores for invalid completions — a property that is precisely the source of masking's advantage in our analysis. For this reason, importing their framework would not yield a clean absorbing-vs-uniform comparison. We view the two approaches as addressing different questions: Wakasugi & Suzuki (2025) reduce absolute dimension dependence for certain reversible kernels under strong geometric assumptions, while we isolate the effect of kernel choice in the original discrete space under minimal assumptions. We will clarify this positioning in the revision and highlight the Section 4 refinements more prominently. Extending embedding-based analyses to non-reversible kernels with zero scores is an interesting open question.
>
>
> ***W2: Limited technical novelty; viewed as natural extension of minimax discrete distribution estimation. The paper would benefit from highlighting any unique technical hurdles overcome specifically for the diffusion trajectory.***
>
> We appreciate this suggestion and will better highlight the unique technical challenges in the revision. The key challenge beyond standard discrete distribution estimation is that discrete diffusion requires controlling score estimation error **simultaneously across a continuous-time trajectory** $t \in [t_0, T]$, where the support, score range, and transition structure of $p_t$ all evolve with $t$. In contrast, standard minimax arguments cannot capture the trajectory-dependent phenomena that drive the kernel separation. Specifically:
>
> (1) **Score range charaterization.** Deriving the closed-form score ranges (Lemmas 3.4–3.5) requires exploiting the conditional independence structure specific to each kernel. The resulting asymmetry—$C_t^{\text{absorb}}$ decays to zero while $C_t^{\text{unif}}$ stays bounded away from zero—drives the sample complexity gap and is invisible to generic minimax arguments.
>
> (2) **Support evolution along the trajectory.** Bounding the cumulative complexity $I_{[t_0,T]}$ requires tracking how each forward kernel reshapes the support of $p_t$ along its corruption path (Lemmas 3.6–3.7). This support evolution analysis—showing absorbing preserves sparsity while uniform expands to $K^d$—has no analogue in static estimation.
>
> (3) **Structurally distinct adaptivity proofs.** The adaptivity results (Section 4) require structurally distinct proof strategies per kernel: decomposition via a "significant sequence set" $\mathcal{E}$ for uniform, and exploiting sparse valid unmasking transitions for the $K_{\max}$ bound in absorbing. These reflect fundamentally different kernel–data interactions.
>
> ***W3: Missing related work: Liang et al. (NeurIPS 2025), Huang et al. (arXiv 2509.21835).***
>
> We thank the reviewer for these references. We cited Liang et al. (2025) in our related work but did not discuss it in detail; we will expand the discussion in the revision. We will also add Huang et al. (2025). Both works study sampling complexity (how many score evaluations at inference time), which is complementary to our sample complexity focus (how many training samples for score estimation). These perspectives are complementary: their results show masking helps at inference time, ours show masking helps at training time.
>
> We hope these responses address your concerns and are happy to discuss further.

---

> > ### Author Rebuttal · Reviewer_4cnz · 2026-04-03
> >
> > Thank you for your response. I believe most of my concerns have been adequately addressed. However, my novelty concern remains partially unresolved. Specifically, the first technical novelty claimed by the authors — the score range characterization — already appears explicitly in the proofs of [2] and [3]. While those works primarily focus on inference complexity, the score range characterization is nonetheless an established component within them. That said, I do acknowledge the contribution of the support evaluation along the generation trajectory as a meaningful addition. Considering the above, I have raised my rating to 4.

---

> > > ### Author Response · Authors · 2026-04-04
> > >
> > > Thank you for raising your score! To clarify: in our ***W2*** response, the three arguments were illustrating what makes score estimation in discrete diffusion technically different from standard static discrete distribution estimation — not claiming that these characterizations are all new relative to the sampling complexity literature. We appreciate pointing out the connection to [2] and [3], and will acknowledge this shared structure explicitly in the revision. We will also make it more clear that the support evolution (Lemmas 3.6–3.7) and adaptivity analyses (Section 4) are our key **technical** contributions.

---

### Official Review · Reviewer_sTB1 · 2026-03-18

**Soundness:** 3
**Presentation:** 3
**Significance:** 3
**Originality:** 3
**Overall Recommendation:** 4
**Confidence:** 4

**Summary:**

Discrete diffusion models are gaining increasing popularity, and a key design choice is whether to use uniform-state or masked diffusion. This paper develops the statistical theory of generalization for both approaches and establishes corresponding generalization bounds. The main takeaway is that masked diffusion is preferable when the data distribution is concentrated on a narrow support, whereas uniform-state diffusion is preferable otherwise.

**Compliance With Llm Reviewing Policy:**

Affirmed.

**Key Questions For Authors:**

can the authors explain the observation that uniform-state diffusion performs worse in terms of likelihood, yet outperforms masked diffusion and autoregressive (AR) models on reasoning benchmarks? [1]


[1] Sahoo et al., 2026 "Scaling beyond masked diffusion language models"

**Limitations:**

Yes

**Strengths And Weaknesses:**

Strengths

The analysis is rigorous and valuable to the community, especially as uniform-state diffusion models are gaining traction [1, 2].

Weaknesses

The empirical evidence is limited to experiments at a very small scale. I would have liked to see this analysis validated on language generation tasks as well. For instance, can the authors explain the observation that uniform-state diffusion performs worse in terms of likelihood, yet outperforms masked diffusion and autoregressive (AR) models on reasoning benchmarks? [1]

Comments

Please consider citing Duo [2] and Duo++ [3], the state-of-the-art uniform-state diffusion method.

### References

[1] Sahoo et al., 2026 "Scaling beyond masked diffusion language models"

[2] Sahoo et al., ICML 2025 "The Diffusion Duality"

[3] Deschenaux et al., ICML 2025 "The Diffusion Duality, chapter 2: Psi Samplers and Efficient Curriculum"

---

> ### Author Rebuttal · Authors · 2026-03-31
>
> Thank you for the positive assessment and for the recognition of the rigor and timeliness of our analysis, particularly in the context of increasing interest in uniform-state diffusion models. We address the reviewer's question and suggestions below.
>
> ***Q: Why does uniform-state diffusion have worse likelihood yet outperform masked diffusion/AR on reasoning benchmarks (GSM8K) in Sahoo et al. (2026)?***
>
> Our analysis compares masking and uniform processes specifically in terms of statistical estimation of the data distribution. Specifically, our analysis bounds the integrated score entropy $\int_{t_0}^T L_t \mathrm{d} t$, which is closely related to likelihood/perplexity. On the likelihood side, it has been empirically observed that masked diffusion achieves better perplexity than uniform-state diffusion (Austin et al., 2023; Lou et al. 2024; Sahoo et al., 2024; Sahoo et al., 2026 [1]). Our theory provides one principled explanation for this empirical trend: absorbing (masking) processes preserve the sparsity structure of the data support during corruption, whereas uniform processes spread probability mass across the entire ambient space, leading to a fundamentally larger estimation problem.
>
> The GSM8K finding concerns a different setting: task-specific quality after supervised fine-tuning on an augmented dataset, where sampling dynamics (e.g., uniform's bidirectional self-correction vs. absorbing's fixed unmasking) play a significant role beyond distributional estimation. These sampling-time advantages are orthogonal to the training-time statistical efficiency that our theory characterizes. Notably, on the zero-shot likelihood-based benchmarks (ARC-e, BoolQ, PIQA, etc.) in [1], which are more directly related to our theoretical setting, masked diffusion remains competitive. We will discuss this nuance in the revision.
>
> ***Weakness: Empirical evidence limited to small scale; language generation validation desired.***
>
> We agree that the empirical section is intentionally limited in scale. it allows precise control over the distributional parameters (support size, transition sparsity, $p_0^{\min}$) that our theory identifies as critical. A full LLM setting would confound the comparison with architecture, optimization, and decoding choices, making it difficult to attribute observed behavior to the forward process alone. Meanwhile, much of the existing large-scale literature (He et al., 2023; Sahoo et al., 2024; Nie et al., 2025; Ye et al., 2025) confirms that masked diffusion achieves superior likelihood in practice, consistent with our predictions. We will clarify this experimental scope more explicitly in the revision.
>
> Thank you as well for suggesting the related references on uniform-state diffusion (including Duo [2] and Duo++ [3]). We will add these papers and discuss them in the revised related-work section.

---

> > ### Author Rebuttal · Reviewer_sTB1 · 2026-04-03
> >
> > This is a good paper which is why I retain my positive score.
> >
> > However,
> > 1. The work can't quite explain the observations made in "Scaling beyond MDLM" paper that USDMs can beat MDLM on some downstream task which isn't a big deal breaker by any means.
> >
> > 2. On the empirical side, the authors could have run experiments / analysis on uniform-state diffusion LLMs to make the paper  more compelling.

---

> > > ### Author Response · Authors · 2026-04-04
> > >
> > > Thank you for the positive assessment! We agree that the USDM-vs-MDLM gap on GSM8K is an interesting phenomenon that involves sampling-time dynamics beyond the training-time statistical efficiency we characterize. In the revision, we will strengthen the empirical narrative by more explicitly connecting our theoretical predictions to the large-scale findings reported in the literature, e.g., the consistent perplexity advantage of masking across SEDD, LLaDA, and Sahoo et al. 2026.

---

### Decision · Program_Chairs · 2026-04-30

**Decision:**

Accept (regular)

**Comment:**

This paper provides a rigorous statistical framework comparing masked and uniform forward kernels in discrete diffusion, showing that masking enjoys a generalization advantage by scaling with the effective data support rather than the full state space. All four reviewers found the theoretical contributions sound and the insights valuable, with scores ranging from Weak Accept to Strong Accept. Concerns were raised regarding bound tightness, limited empirical scale, and the constant noise schedule assumption. The authors addressed these effectively during the rebuttal, and reviewers acknowledged resolution of most issues. The paper offers clear, principled explanations for an important empirical phenomenon. I recommend acceptance.